# Assessment of Scalable Fractionation Methodologies to Produce Concentrated Lauric Acid from Black Soldier Fly (*Hermetia illucens*) Larvae Fat

**DOI:** 10.3390/insects16020171

**Published:** 2025-02-06

**Authors:** Luis Vázquez, Carlota Reyero, Raúl Hurtado-Ribeira, David Villanueva-Bermejo, Alejandro Belinchón, José Palomar, Tiziana Fornari, Diana Martín

**Affiliations:** 1Sección Departamental de Ciencias de la Alimentación, Facultad de Ciencias, Universidad Autónoma de Madrid, 28049 Madrid, Spain; carlotareyero@gmail.com (C.R.); raul.hurtador@estudiante.uam.es (R.H.-R.); david.villanueva@uam.es (D.V.-B.); tiziana.fornari@uam.es (T.F.); diana.martin@uam.es (D.M.); 2Departamento de Producción y Caracterización de Nuevos Alimentos, Instituto de Investigación en Ciencias de la Alimentación (CIAL) (CSIC–UAM), 28049 Madrid, Spain; 3Departamento de Ingeniería Química, Facultad de Ciencias, Universidad Autónoma de Madrid, 28049 Madrid, Spain; alejandro.belinchon@uam.es (A.B.); pepe.palomar@uam.es (J.P.)

**Keywords:** distillation, edible insects, fatty acid, *Hermetia illucens*, lauric acid, supercritical fluid extraction (SFE), winterization

## Abstract

The interest in insects as the food of the future is growing due to their sustainability and nutritional value. In particular, the fat content of *Hermetia illucens* larvae stands out for its high lauric acid content, a compound with antimicrobial properties and multifunctional activity. This study explored various fractionation methodologies to concentrate lauric acid and maximize its potential. The versatility and sustainable origin of the resulting products make them highly promising resources.

## 1. Introduction

Nowadays, various sectors, including the food industry, are transitioning towards a circular economy that emphasizes the valorisation of resources, materials, and products. Guided by sustainable development principles, this shift aims to reduce waste accumulation, with a growing focus on utilizing by-products, co-products, and exploring all the valuable ingredients derived from a process. In this context, the fat derived from the larvae of the edible insect *Hermetia illucens* (black soldier fly larvae, BSFL), is a co-product derived from the main process of protein meal production, making it attractive due to its high lauric acid content, which accounts for approximately 50% of the total fatty acids [1,2,3]. Lauric acid is a saturated fatty acid containing twelve carbon atoms, often classified as a medium-chain fatty acid (MCFA) [4]. This fatty acid naturally occurs as part of triacylglycerols in sources such as coconut oil, palm kernel oil, or relatively new alternatives like BSFL. Lauric acid exhibits various beneficial properties, leading to its extensive use in industries such as pharmaceuticals, cosmetics, animal feed, and food products [5,6,7]. Among these beneficial properties, its notable antibacterial activity stands out, which is particularly associated with its free form and its monoglyceride derivative, glycerol monolaurate [8,9,10]. Moreover, lauric acid can be utilized as a precursor for various molecules in different industries, including emulsifiers, medium-chain triacylglycerols, or structured lipids [11]. Concerning its metabolism, lauric acid is rapidly transported through the portal vein to hepatocytes, where it directly enters the mitochondria through passive diffusion, undergoing rapid oxidation. Consequently, this process leads to less fat accumulation in adipose tissue [12]. Additionally, some studies have suggested that lauric acid has the potential to decrease total cholesterol and increase HDL cholesterol levels, demonstrating possible cardioprotective effects [11,12]. In animal feeding, interest in lauric acid has increased in recent years, as a growth enhancer, and due to its immunomodulatory and protective effects at the gastrointestinal level, which are related to microbiota modulation [7,13].

With the aim of maximizing the full potential of the BSFL fat derived from the properties of this fatty acid and enhancing its added value, the concentration of lauric acid becomes interesting and it has not yet been explored. In general, the processes for the production of high-purity specific fatty acid concentrates require two sequential stages: (i) breakdown of fatty acids from the starting triacylglycerols in their natural sources, and (ii) enrichment, concentration, or isolation processes. The liberation of fatty acids from raw fats and oils involves methodologies such as saponification, hydrolysis, and alcoholysis, accomplished through either chemical or enzymatic procedures [14,15,16]. The subsequent fractionation of fatty acids, whether in their free, methyl, or ethyl ester form, can be conducted using various methodologies [17,18,19,20,21,22].

Distillation can be considered the most common and efficient method for producing high-purity fatty acids [17]. The separation of fatty acids by distillation is based on their distinct boiling points, which are associated with their chain length [23]. The feedstock is preheated and enters the distillation column and is quickly vaporized by flowing across heated trays to fractionate fatty acids. The development of molecular distillation, which involves a high vacuum and a short exposure of the sample to high temperatures, has led to improved column efficiencies and may be more suitable for avoiding degradation [24,25]. As an example, Vázquez and Akoh [26] used this technique to effectively fractionate short and medium-chain fatty acid ethyl esters from a blend of coconut oil and dairy fat.

In addition to distillation, winterization and supercritical fluid extraction hold special significance as they can be cost-effective, align with Green Chemistry, and are feasible for scaling in industrial applications. In recent years, the combination of technologies classified as ‘green’, such as enzymatic pretreatments, ultrasound, pressurized liquid extraction, microwave, as well as SFE and winterization, has undergone significant advancements for the extraction and fractionation of lipid components from different matrices [27,28,29]. Furthermore, there is currently a growing trend in the use of non-traditional, bio-based, or food-grade solvents in these applications, such as limonene [30].

Winterization involves the partial crystallization of the fatty acids at low temperatures, leading to the separation of solids from the liquid fraction. The melting points of fatty acids vary depending on unsaturation levels, enabling the separation of mixtures of saturated and unsaturated fatty acids by winterization, but the proper choice of solvent and temperature is required to effectively concentrate specific fatty acids [19,31].

Supercritical fluid extraction (SFE) has advanced significantly since its inception and is widely recognized as a clean and environmentally friendly ‘green’ processing technique [27,32,33]. Carbon dioxide is the most acceptable supercritical solvent in food applications because of its low critical temperature and pressure, low cost, wide availability, and non-flammability [34]. Due to the non-polar nature of CO_2_, considerable interest has been shown in the oil and fat industry [35,36,37]. In this field, the fractionation of fatty acids is based on their distinct chain lengths, leading to varying molecular weights and, consequently, different solubilities in supercritical CO_2_ (sc-CO_2_). Some studies have reported the fractionation of fats and oils into triacylglycerol fractions with varying properties [38,39]. Furthermore, fatty acids released in their ester form are more stable than their corresponding free fatty acids and are preferred for supercritical fractionation due to their higher solubility in dense CO_2_ [40]. To the best of our knowledge, the use of sc-CO_2_ has been limited, and winterization has not been explored for the fractionation of lauric acid. Building upon those premises, the objective of the present study was to evaluate the potential of winterization and sc-CO_2_ extraction to produce lauric acid concentrates from BSFL fat. Additionally, the experimental results obtained were compared with flash and multistage distillation process simulation, a more conventional method of concentration of fatty acids.

## 2. Materials and Methods

### 2.1. Raw Materials and Chemicals

Dried BSFL were provided by Entomo Agroindustrial (Cehegín, Murcia, Spain) after slaughtering by blanching and drying by oven-drying, as described by Hurtado-Ribeira et al. [3]. The BSFL were fed a conventional diet based on wheat bran. The fat was extracted by mechanical pressing, following the procedure of Hurtado-Ribeira et al. [3]. Potassium hydroxide, ethanol (96%) (volume/volume, *v*/*v*), and absolute ethanol were obtained from Scharlab S.L. (Barcelona, Spain). Hydrochloric acid (37%), sodium sulphate anhydrous, lauric acid (98%), methyl laurate (98%), sodium hydroxide, boron trifluoride (BF_3_, 14% *v*/*v* in methanol), sodium ethoxide (95%), and sodium ethoxide solution (21%) (weight/volume, *w*/*v*) in ethanol were purchased from Merck KGaA (Darmstadt, Germany). Potassium hydrogen phthalate was acquired from Panreac Química S.L.U. (Barcelona, Spain). Hexane (95%), acetone, and methyl alcohol anhydrous, all HPLC grade solvents, were purchased from Macron Fine Chemicals™ (Center Valley, PA, USA).

### 2.2. Production of Fatty Acids in Free and Ethyl Ester Forms

#### 2.2.1. Chemical Hydrolysis

Saponification was carried out to hydrolyse the triacylglycerols of BSFL fat into their corresponding free fatty acids (FFAs), following the procedure by Vázquez et al. [16]. Firstly, 50 g of BSFL fat were melted at 50 °C, and dissolved into 80 mL of potassium hydroxide 3.7 N in 96% water–ethanol (1:1, *v*/*v*). The mixture was maintained at 50 °C using an IKA RCT basic temperature control system (Staufen, Germany), with mechanical agitation by a magnetic stirrer. After 20 min, the reaction was stopped by adding 40 mL of distilled water, followed by 65 mL of 4 M hydrochloric acid to lower the pH to 2 and release the FFA. Subsequently, the FFAs were separated by decantation, followed by a washing step using 20 mL of distilled water. Finally, this product was dried over anhydrous sodium sulphate and filtered under vacuum. The chemical hydrolysis of BSFL fat was replicated and the product was used as starting material for subsequent winterization processes.

#### 2.2.2. Chemical Ethanolysis

Transesterification reactions of BSFL fat were carried out to convert the triacylglycerols into their corresponding fatty acid ethyl esters (FAEEs). A total of 500 g of BSFL fat was mixed with a solution of solid sodium ethoxide in absolute ethanol at a concentration of 8% (weight/weight, *w*/*w*) in a Kiloclave type reactor (Buchi, Uster, Switzerland), using a double-jacketed tank of 1 L volume with mechanical stirring, and coupled to a thermostat (F32-HE, Julabo, Seelbach, Germany). The reaction was carried out at 60 °C for 40 min with mechanical agitation at 200 rpm. The product was centrifuged at 9000 rpm for 15 min at 40 °C (Thermo Fisher Scientific Sorvall LYNX 6000, Waltham, MA, USA). The bottom phase, which contained solid impurities and glycerine, was removed, recovering the organic phase. The organic phase was washed using 500 mL of distilled water. It was then centrifuged at 11,200 rpm for 10 min at 25 °C, recovering the upper organic phase composed of FAEEs. The final product was dried with anhydrous sodium sulphate and vacuum filtered. The chemical ethanolysis was replicated and the product was used as starting material for subsequent sc-CO_2_ fractionation.

### 2.3. Concentration of Lauric Acid

#### 2.3.1. Winterization of FFA

Low-temperature crystallization procedures were based on the methodology described by Vázquez and Akoh [19], primarily employing hexane as solvent. The FFAs obtained from chemical hydrolysis were dissolved into solvents, homogenised by vortex, and subsequently stored at −20 °C for 24 h. The FFA to hexane ratios studied were 1:4, 1:6, 1:8, 1:10, 1:15, and 1:20 weight/volume (*w*/*v*) (processes named as P1 to P6, respectively). At the optimal FFA to solvent ratio, two additional experiments were conducted using absolute ethanol and acetone as solvents for comparison. After winterization, the samples were centrifuged at 15,000 rpm for 2 min at −20 °C (Thermo Fisher Scientific Sorvall LYNX 6000, Waltham, MA, USA), to separate a liquid phase enriched in monounsaturated (MUFAs) and polyunsaturated fatty acids (PUFAs), and a solid phase primarily composed of saturated fatty acids (SFAs). Liquid and solid fractions were dried in a rotary evaporator (BUCHI Ibérica S.L.U., Barcelona, Spain), and subsequently analysed by gas chromatography. Winterization processes were carried out at least in duplicate. Two main responses were evaluated throughout the study, defined as follows:

Concentration or purity: weight percentage (wt%) of a single fatty acid in the composition, expressed in Equation (1):(1) Fatty acid weight in a fraction (mg)Weight of the whole fraction (mg)×100

Recovery: % of one fatty acid regarding its amount in the starting material, expressed in Equation (2):(2)                    Fatty acid weight in a fraction (mg)Weight of fatty acid in starting material (mg)×100

#### 2.3.2. Supercritical Fluid Extraction of FAEE

A semi-pilot scale supercritical fluid extractor (model: SF2000, Thar Technology, Pittsburgh, PA, USA) was used for the fractionation of ethyl laurate from the FAEE mixture obtained by the chemical ethanolysis of BSFL fat (Figure 1). The experimental device comprises a counter-current extraction column (280 cm total height) with two levels of the feed inlet (oil sample) and two separators (500 mL each) where the decompression and recovery of the extract takes place (Figure 1). Each section of the packed column and the separators has independent control of temperature (±2 °C). The extraction column (316 stainless) has an internal diameter of 2.97 cm, and it is packed with a structured material (17-4PH H1150). The effective packed height (from the feed to the CO_2_ inlet) was 120 cm. The extraction unit also includes a recirculation system, where CO_2_ is condensed, pumped, and heated up to the desired pressure and temperature. The pressure in the extraction column is automatically controlled by a back pressure regulator valve (±0.1 MPa). The CO_2_ flow rate is measured using a flow meter from Siemens AIS (model: Sitrans FC Mass 2100 DI 1.5, Nordborgvej, Denmark).

Supercritical CO_2_ was introduced into the column from the bottom and once the operating pressure and temperature were reached, the product obtained from BSFL ethanolysis (FAEE mixture) was pumped from the top at a constant flow rate of 2.9 g/min (approximately 200 mL/h) during 60 min. BSFL ethanolysis instead of hydrolysis was selected to produce the starting SFE material because (i) FAEEs are in a liquid state, easily pumpable, and thus capable of flowing through the extraction column, unlike FFAs, which are in a solid state at ambient temperature; and (ii) FAEEs are less polar than FFAs, making them more soluble in supercritical CO_2_, allowing the process to be carried out under milder conditions. FAEEs were selected instead of fatty acid methyl esters (FAMEs) to avoid the use of methanol, which is highly toxic and not permitted in the food industry. SFE processes were carried out at a temperature of 55 °C, pressure of 115 bar, and CO_2_ flow rate of 70 g/min, according to preliminary experiments. Once the extraction finished, sc-CO_2_ was pumped for another 20 min to complete the extraction of the remaining FAEEs inside the column. Products were collected from the top (extract) and the bottom (raffinate) of the column. The extract was collected from the first separator by depressurization at 20 bar and 40 °C. SFE processes with FAEEs from BSFL fat were carried out at least in duplicate. As in winterization, concentration (i.e., purity) and recovery were evaluated as the main responses of the SFE study.

#### 2.3.3. Simulation of Distillation

Distillation evaluation has been performed by using the professional process simulator Aspen Plus V14 to model the separation of the lauric compound from a benchmark initial mixture with the fatty acid profile of BSFL, considering two distinct chemical structures: (i) mixture formed by free fatty acids; and (ii) mixture formed by the corresponding fatty acids in the form of ethyl ester (Table 1).

The COSMO/Aspen methodology [41] was applied for process simulations. For this purpose, Turbomole 4.5.2. software was used to optimize the molecular geometries of all acids and ethyl ester compounds by the DFT method, listed in Table 1, at the BP86/TZPV computational level using the COSMO continuum solvation method [42]. Subsequently, COSMOThermX19 software was applied to predict different molecular and thermodynamic properties (σ-profile, molar weight, density, boiling point, and COSMO volume) of the compounds, by using the COSMO-RS method with the BP_TZVP_19 parametrization. The final step in the COSMO/Aspen strategy involved employing Aspen Plus 14 software to model the simulated separation process, by using the COSMO-SAC thermodynamic model. Those acid and ethyl ester compounds not yet present in the Aspen Plus databank were introduced as pseudocomponents using the COSMO/Aspen-based methodology detailed elsewhere [42]. The objective of the distillation process is to recover the C12:0 compound (lauric acid or ethyl laurate) from a mixture with a fixed flow of 10,000 kg/h and an initial composition, described in Table 1. Two distillation approaches are considered for the separation of the C12:0 compound.

Simple one-stage flash distillation (Figure 2A). In this case, the FLASH module of the Aspen Plus process simulator was utilized to simulate the separation process. Wide ranges of operating temperature (100–240 °C) and pressure (10-4-1 bar) were studied to analyse separation efficiency, i.e., the purity and recovery of the C12:0 compound in the vapor stream.Multistage distillation column (Figure 2B). The RADFRAC module of the Aspen Plus process simulator was employed in equilibrium mode for distillation column modelling, using 10 stages. A total condenser and a kettle reboiler were incorporated with a reflux ratio set to 1. The distillate flow rate from the column was adjusted to ensure an 80% recovery of the C12:0 compound, and the impact of the column’s operating pressure on the reboiler temperature and distillate purity of the C12:0 compound was investigated.

### 2.4. Analysis by Gas Chromatography

Prior to analysis by GC, FFAs obtained from winterization processes were transformed into fatty acid methyl esters (FAMEs). This methylation was carried out according to the Association of Official Agricultural Chemists (AOAC) Official Method 996.01 (Section E), using a NaOH–methanol solution (0.5 N) and BF3–methanol solution (~14%, *w*/*v*) as catalysts [43]. These FAMEs, and the FAEEs obtained from SFE processes, were dissolved in hexane at a concentration of 8 mg/mL, and further analysed by GC according to the method described by Vázquez et al. [44]. The identification and quantification of FAMEs and FAEEs were carried out in an Agilent 6850 Network GC System (Avondale, AZ, USA), coupled to an FID detector and Agilent 6850 autosampler. The capillary column was an HP-88 (30 m, 0.25 mm i.d.) (Avondale, AZ, USA). An injection volume of 1 μL and a 20:1 split ratio were used. The injector and detector temperatures were 220 and 250 °C, respectively. The temperature program started at 50 °C, rising to 220 °C at 15 °C min^−1^. The final temperature, 220 °C, was held for 10 min. The identification of FAMEs and FAEEs was based on retention times and the relative area percentages as compared to the No.3 PUFA reference standard (47085-U), obtained from Supelco (Bellefonte, PA, USA). For quantification, the external standard methodology with a calibration curve of methyl laurate was utilized.

### 2.5. Peroxide Value Determination

To evaluate the impact of the processes on lipid oxidation, a key attribute of lipid quality, the peroxide value (PV) of products obtained from winterization and SFE processes was determined by duplicate. The photometric FoodLab instrument (CDR S.r.L., Ginestra Fiorentina, Firenze, Italy) was used, following the method outlined by Martín et al. [45]. This method, equivalent to the American Oil Chemists’ Society (AOCS) official method, Cd 8–53, is based on a rapid photometric reaction of 5 µL of the sample, added by a positive displacement pipette to 1 mL of the reaction reagent of the supplier in prefilled cuvettes. Then, 10 µL of the second reagent of the supplier was added. The mixture was reacted for 3 min at 37 °C in the incubation cells of the thermostated instrument. The samples were then measured at 505 nm in the measuring cells, with a measuring range for quantification given from 0.1 to 50 mEqO_2_/kg, using the default settings for calibration and quantification.

### 2.6. Statistical Analysis

Statistical differences among the studied conditions were assessed using a one-way analysis of variance (ANOVA) conducted with the general linear model procedure in the SPSS 26.0 statistical package (SPSS Inc., Chicago, IL, USA). When the effect of condition was significant (*p* ≤ 0.05), differences between groups were analysed using post hoc Tukey’s tests.

## 3. Results and Discussion

### 3.1. Lauric Acid Concentration via Winterization

The FFAs obtained through chemical hydrolysis of BFSL fat were used as the starting material for the winterization processes. The FFA content in the final product was approximately 99%, according to the GC analysis. Firstly, the effect of the oil-to-solvent ratio (*w*/*v*) on the concentration (Table 2) and recovery (Table 3) of lauric acid in hexane was investigated.

As observed in Table 2, lauric acid, which initially was present at 50% in raw BSFL, was concentrated in the solid fraction in all processes (from 57% up to 65%). Unsaturated fatty acids (oleic acid, linoleic acid, and linolenic acid) could be dissolved more effectively in the solvent, causing lauric acid to solidify and concentrate predominantly in the solid fraction. Statistical differences were observed between P1, P2, and P3 in comparison to P4, P5, and P6, demonstrating a significant increase (*p* < 0.05) in lauric acid concentration from the sample-to-solvent ratio of 1:10 onwards. However, beyond this value, the concentration of lauric acid did not occur as the amount of solvent increased.

It should be highlighted that using a sample-to-solvent ratio of 1:10 achieves an enrichment of approximately 30% (64.6%) of lauric acid compared to the starting material (50.3%).

Table 3 shows that saturated fatty acids (lauric acid, myristic acid, palmitic acid, and stearic acid) were predominantly recovered in the solid fraction (SF) in all cases, whereas unsaturated fatty acids had a greater tendency to be recovered in the liquid fraction (LF), especially when the oil-to-solvent ratio increased. In addition, significant differences (*p* < 0.05) in the recovery of lauric acid were observed among the different solid fractions. A strong negative linear relationship (R^2^ = 0.985) was observed with the proportion of solvent increase, indicating that the higher the amount of solvent employed, the lower the recovery of lauric acid. Hence, in the process with the least amount of solvent (P1), a recovery of ~91% of lauric acid was achieved, while in P6, a recovery of ~53% was obtained. This is because greater amounts of solvent could be able to dissolve higher amounts of lauric acid, which went on to be more easily recovered in the liquid fraction, as observed in Table 3 (from 13% in P1 to 54% in P6).

Therefore, considering both the concentration and recovery of lauric acid, simultaneously, it was concluded that the optimal conditions correspond to process P4. Thus, the 1:10 oil-to-solvent ratio (*w*/*v*) provided a high concentration of lauric acid (~65%), with a recovery of ~81% in the solid fraction, indicating only a small loss of ~19% in the liquid fraction. Additionally, the use of a lower amount of solvent for P4 compared to P5 and P6 represents a significant cost advantage that may make potential industrial scaling more feasible. In addition, it is interesting to remark that the P4 condition would lead to the production of the LF as a co-product with a FFA profile that is also of interest compared to the raw BSFL fat. This LF contained an important decrease in saturated fatty acids, but still contained more than 30% of the target lauric acid, together with a relevant increase in unsaturated fatty acids.

Subsequently, the oxidation of solid and liquid fractions from the previously described processes (after solvent removal) was determined (see Table 4). It should be taken into account that the peroxide value of the FFAs used as the starting material was 1.01 ± 0.13 mEq O_2_/kg.

As observed in Table 4, the liquid fractions showed higher oxidation compared to the solid fractions in the processes that utilized a reduced amount of the solvent (P1, P2, and P3). This fact can be explained because unsaturated fatty acids are predominantly present in liquid fractions, and due to their double bonds, they have a higher tendency to undergo oxidation. It should also be noted that similar values were obtained across different processes, with no statistical differences in the solid fraction (*p* > 0.05). Considering the peroxide value of the starting FFAs, it can be concluded that winterization is a process in which the target product (solid fractions) exhibited minimal oxidation (mean value 1.21 mEq O_2_/kg), showing no statistical differences with the starting material (*p* > 0.05).

In addition to hexane, other solvents such as ethanol and acetone are permitted in the food industry. In this context, it was interesting to study their behaviour in winterization processes for concentrating lauric acid. Table 5 presents the results obtained using these solvents at a 1:10 oil-to-solvent ratio (*w*/*v*).

As observed in Table 5, statistical differences (*p* < 0.05) were noted in the concentration of lauric acid among the different solvents used. It was observed that acetone was not effective, yielding a significantly lower value (56.3%) compared to processes using hexane (64.6%) and ethanol (59.1%). The results with acetone are consistent with those previously obtained by Vázquez and Akoh [46], where acetone showed good results in separating TAG with different fatty acid profiles but was not effective in the fractionation of FFAs [19,46].

In addition, a noticeable result was observed when using ethanol as a solvent, since in this case, lauric acid was predominantly recovered in the liquid phase (~84%). Hence, in contrast to what happens with hexane, where lauric acid precipitates in the solid fraction along with other saturated acids such as myristic and palmitic acids, with ethanol, due to the change in polarity, lauric acid increases its solubility and is recovered in the liquid fraction along with unsaturated fatty acids. Therefore, although the best option for concentrating lauric acid through winterization was by using hexane with an oil-to-solvent ratio of 1:10 (*w*/*v*), alternatively, the winterization process with ethanol can be also considered an interesting method, as it yields a good concentration (59.1%), with a 17% enrichment with respect to the raw material, and high recovery (~84%). Furthermore, it yielded a product with a more nutritionally favourable fatty acid composition compared to the SF, exhibiting significantly higher levels of MUFAs and PUFAs, and a lower content of palmitic acid (*p* < 0.05%).

### 3.2. Lauric Acid Concentration via Supercritical Fluid Extraction

Based on previous studies, the conditions of 115 bar, 55 °C, and 70 g/min of CO_2_ mass flow rate were used to investigate the concentration of lauric acid in FAEEs derived from BSFL fat. The FAEE content in the final product after ethanolysis was approximately 99%, according to the GC analysis.

The mass of fat collected as extract and raffinate were 86.3 g and 67.5 g. Considering the total mass of products recovered (sum of the mass of extract and raffinate) in relation to the mass of feed pumped (172 g), a mass balance with an accuracy of around 90% was reached in the experiments. Approximately 10% of the material remained as a liquid, impregnating the packing within the extraction column. As shown in Figure 3, the process was highly effective in concentrating lauric acid, increasing it from ~50% in the initial BSFL fat to ~80% in the extract. In addition, it is worth mentioning the high recovery of lauric acid, reaching around 85%. Thus, it was confirmed that the selected combination of parameters—temperature (55 °C) and pressure (115 bar)—resulted in a CO_2_ density with a sufficient solvent capacity to effectively and selectively solubilize lauric acid. Furthermore, the solvent-to-feed ratio employed ensured an adequate amount of CO_2_ to extract this compound with high recovery, which constitutes a significant proportion of the starting material.

Concerning other fatty acids, it was observed that myristic acid was distributed approximately evenly between the separator and the refined product, resulting in an 8.5% concentration in the extract. Moreover, only 20% of palmitic acid was recovered in the extract, leading to a concentration of 4%. The remaining major fatty acids were mainly recovered in the raffinate, causing their concentration in the extract to be less than 1.5% in all cases.

Additionally, the oxidation state of SFE products was determined, with an initial peroxide value of 2.61 mEq O_2_/kg in the starting FAEEs from BSFL fat. It was observed that the oxidation state was significantly lower (*p* < 0.05) in the extract, enriched in lauric acid, compared to the raffinate, with peroxide values of 0.93 ± 0.07 and 5.15 ± 1.37 mEq O_2_/kg, respectively. The raffinate could exhibit a higher peroxide value due to its increased content of unsaturated FAEEs, which are more prone to oxidation. Additionally, primary oxidation compounds, by capturing active oxygen species, increase their polarity, making them less soluble in CO_2_ and resulting in their enrichment in the raffinate. Hence, the supercritical technology was highly effective in obtaining a product with a 60% enrichment in lauric acid compared to the raw material, while simultaneously improving its oxidation state.

### 3.3. Lauric Acid Concentration via Simulated Distillation

Process simulations using Aspen Plus V14 are highly advanced and reliable. Given that distillation is a widely used technique, this comparison could provide valuable insights. Firstly, one-stage distillation was modelled in a wide range of operating temperatures and pressures to evaluate the separation efficiency of lauric acid from the initial FFA mixture described in Table 1, simulating the BSFL fatty acid profile. Figure 4 presents the obtained mass purity (i.e., concentration) and recovery of lauric acid product. As can be seen, product purity and recovery increases at a lower operating pressure. In contrast, increasing the operating temperature means higher product recovery but lower product purity, and the opposite trend is observed when decreasing the temperature.

Table 6 presents the proposed design of the one-stage distillation to achieve the highest purity (Case 1) or highest recovery (Case 2) of lauric acid.

To improve separation efficiency, the distillation of the FFA mixture was evaluated in a 10-stage column with a reflux ratio of 1, fixing the distillate flow rate from the column to achieve the 80% recovery of lauric acid. Figure 5 shows the impact of the operating pressure on the lauric acid purity and the reboiler temperature. Increasing the vacuum (lower pressure) in the column implies a lower operating temperature in the reboiler, achieving higher purity. Table 6 collects the design of 10-stage distillation proposed in this work to concentrate lauric acid, preserving its thermal stability by operating at 120 °C in the reboiler [47]. As can be seen, multistage distillation allows for obtaining a remarkably high product purity (96.7%) with reasonably high lauric acid recovery (80%), clearly improving the separation efficiency obtained by one-stage distillation.

The above-described process simulation analysis has been also performed for the separation of ethyl laurate from the initial FAEE mixture described in Table 1 (see Appendix A) by one-stage and 10-stage distillation. The proposed distillation designs for the ethyl laurate concentration are collected in Table 7. As can be seen, similar separation efficiencies are obtained in comparison to the free lauric acid case (Table 6) for both one-stage and 10-stage distillations. The main difference is related to the lower boiling point of the ester (271 °C at 1 atm) in comparison to the acid (298.9 °C at 1 atm) compound, which implies the advantage of working at a lower operating temperature in one-stage distillation and with a lower vacuum (higher pressure) in the 10-stage distillation.

### 3.4. Comparison of Results from the Three Fractionation Technologies

As a summary, Figure 6 shows the comparison of the results obtained by the different technologies employed.

As illustrated in Figure 6, SFE represents the most suitable compromise between the concentration (~80%) and recovery (~85%) of lauric acid, as it provided an optimal balance between these responses in comparison to other methodologies evaluated. To improve the lauric acid purity beyond SFE results, a 10-stage distillation would be necessary, providing a concentration of ~97% lauric acid with a recovery of around 80%. Conducting the process through one-stage distillation could yield high purity (86–87%), but the recovery would be very poor (20–26%). Alternatively, winterization can also offer an acceptable concentration of lauric acid (~65%) with good recovery (~81%).

At this point, a further study focused on the costs, energy consumption, and environmental impacts associated with the three technologies would be necessary. Winterization is cost-effective and relatively low in energy consumption, making it suitable for small to medium-scale operations. However, its reliance on solvents raises environmental and safety concerns due to disposal challenges [48]. Supercritical CO_2_ extraction provides good concentration and selectivity, making it ideal for high-value applications, but its initial cost can be high due to the need for advanced equipment, along with CO_2_ consumption, which, while often recyclable, still may have a notable environmental footprint [49]. Vacuum distillation, on the other hand, is highly scalable and efficient for large volumes, with a lower thermal degradation of products due to reduced boiling points under a vacuum. However, its energy demands for maintaining high vacuum conditions and heating are substantial, leading to higher operational costs and potential environmental impacts [50].

## 4. Conclusions

Through this work, it has been demonstrated that winterization and supercritical fluid extraction are effective in concentrating lauric acid from the fat of *Hermetia illucens* larvae. Winterization primarily separated components based on the presence of double bonds, whereas sc-CO_2_ separated components by chain length. Specifically, extraction with sc-CO_2_ resulted in a product with high concentration values of this fatty acid (~80%), also providing high recovery (~85%). Overall, these values could be compared to those theoretically obtained through a multistage distillation process using a 10-stage equilibrium distillation column. Although winterization yielded slightly lower results, it can be a much more cost-effective and simpler alternative for producing lauric acid concentrates. Additionally, the oxidation of the product obtained through either winterization or SFE was minimal, highlighting this process as a viable and effective method for obtaining high-value lipids from edible insects. Nevertheless, selecting the most appropriate technology for concentrating lauric acid would require conducting a comprehensive study to thoroughly assess the economic costs, including equipment investment, energy consumption, and environmental impacts, in addition to evaluate specific challenges associated with each technology.

## Figures and Tables

**Figure 1 insects-16-00171-f001:**
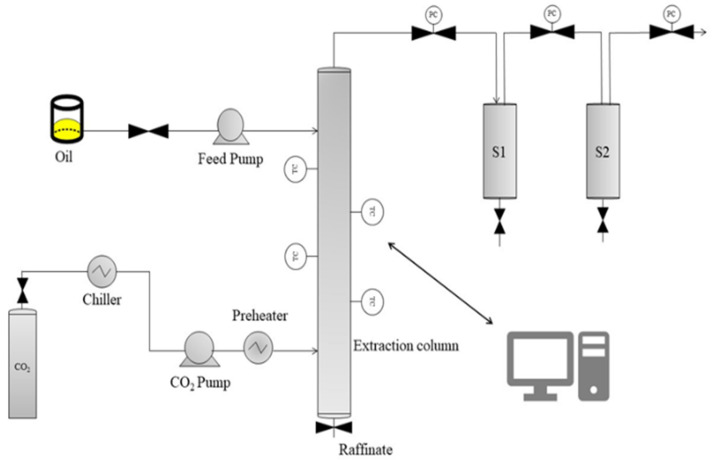
Schematic diagram of the sc-CO_2_ fluid extraction unit. (TC) temperature controller, (PC) pressure controller.

**Figure 2 insects-16-00171-f002:**
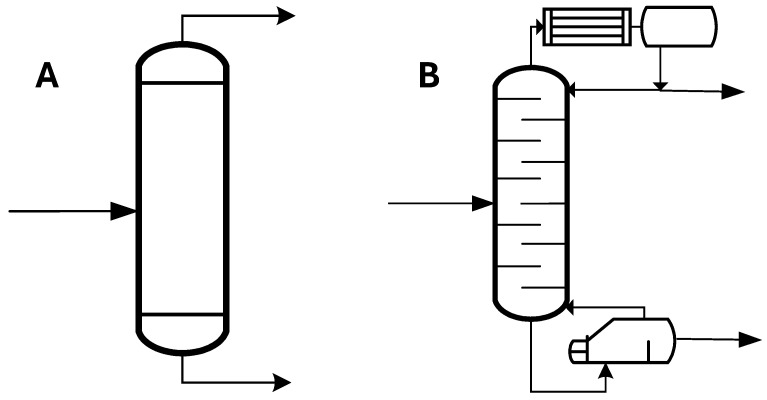
Two alternative distillation process flowsheets to recover the lauric/ethyl laurate compound form the initial mixture stream. (**A**) simple one-stage distillation and (**B**) 10-stage distillate column.

**Figure 3 insects-16-00171-f003:**
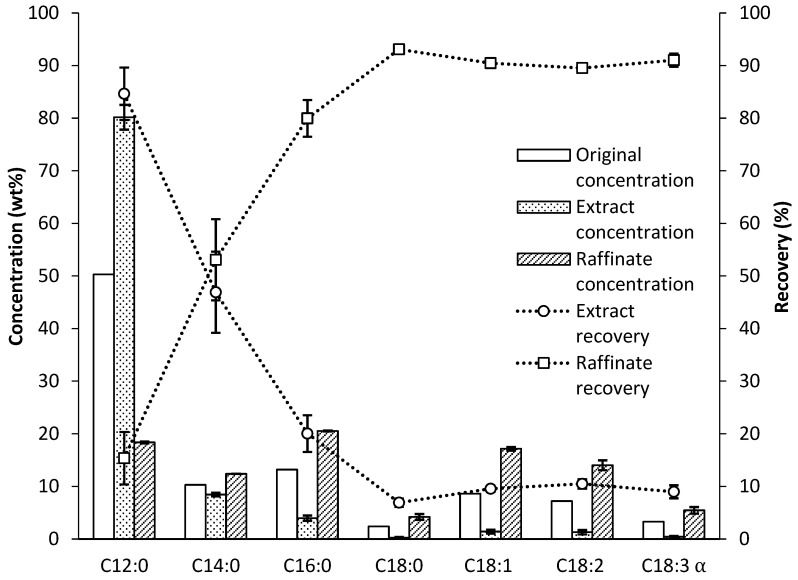
Concentration (wt%) in columns and recovery (%) in lines of SFE processes with FAEEs from BSFL fat.

**Figure 4 insects-16-00171-f004:**
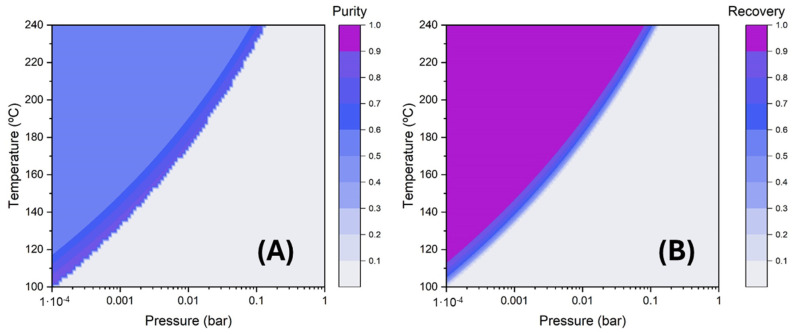
(**A**) Mass purity and (**B**) recovery of lauric acid in the vapor stream employing a simple distillation.

**Figure 5 insects-16-00171-f005:**
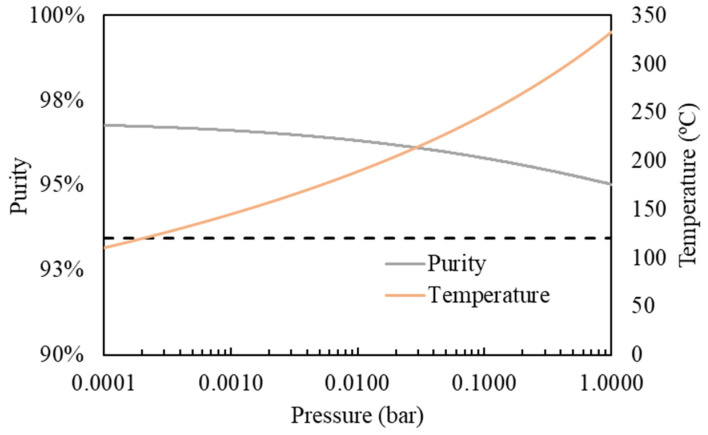
Product mass purity, reboiler temperature, and pressure of the multistage distillation column are maintained with a constant 80% lauric acid recovery.

**Figure 6 insects-16-00171-f006:**
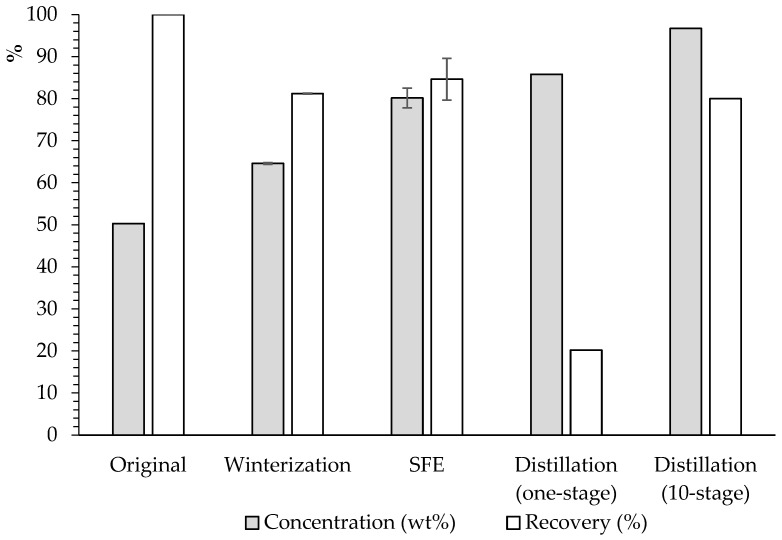
Concentration (wt%) (grey) and recovery (%) (white) of lauric acid by the different technologies assessed. All methodologies used free fatty acid form except SFE, which used ethyl esters.

**Table 1 insects-16-00171-t001:** Mass composition of the FFA/FAEE feed stream to be separated considered in the scenario involving free fatty acids or their corresponding ethyl esters.

Structure		Nomenclature	Mass Fraction (%)
1. Free Fatty Acid	2. Ethyl Ester		
Capric acid	Ethyl Caprate	C10:0	0.9
Lauric acid	Ethyl Laurate	C12:0	50.3
Myristic acid	Ethyl Myristate	C14:0	10.3
Myristoleic acid	Ethyl Myristoleic	C14:1	0.4
Palmitic acid	Ethyl Palmitate	C16:0	13.2
Palmitoleic acid	Ethyl Palmitoleic	C16:1	2.4
Margaric acid	Ethyl Margarate	C17:0	0.4
Heptadecenoic acid	Ethyl Heptadecenoic	C17:1	0.3
Stearic acid	Ethyl Stearate	C18:0	2.4
Oleic acid	Ethyl Oleate	C18:1 n9	8.6
Linoleic acid	Ethyl Linoleate	C18:2	7.2
γ-linolenic acid	Ethyl Linolenate (γ)	C18:3 n6	0.1
α-linolenic acid	Ethyl Linolenate (α)	C18:3 n3	3.3
Gadeloic acid	Ethyl Gadeloic	C20:1	0.2

**Table 2 insects-16-00171-t002:** FFA concentration (wt%) from hexane winterization processes at different oil-to-solvent ratios (*w*/*v*).

FFA	Raw FFA	P1 Hexane 1:4		P2 Hexane 1:6		P3 Hexane 1:8	
		SF	LF	SF	LF	SF	LF
Lauric	50.3	59.4 ± 0.2 ^b^	28.7 ± 0.6 ^v^	57.3 ± 0.2 ^c^	35.6 ± 1.1 ^x^	59.3 ± 0.2 ^b^	35.6 ± 0.4 ^x^
Myristic	10.3	10.9 ± 0.0 ^b^	8.4 ± 0.1 ^x^	10.9 ± 0.0 ^b^	8.3 ± 0.0 ^x^	11.1 ± 0.1 ^a^	8.5 ± 0.1 ^x^
Palmitic	13.2	12.3 ± 0.1^de^	4.0 ± 0.0 ^v^	12.0 ± 0.0 ^e^	5.3 ± 0.4 ^x^	12.7 ± 0.1 ^d^	5.0 ± 0.1 ^xw^
Stearic	2.4	1.7 ± 0.0 ^e^	0.9 ± 0.0 ^x^	1.7 ± 0.0 ^de^	0.9 ± 0.0 ^x^	1.8 ± 0.0 ^cd^	0.9 ± 0.1 ^x^
Oleic	8.6	5.6 ± 0.1 ^b^	21.4 ± 0.1 ^z^	6.5 ± 0.1 ^a^	18.3 ± 0.6 ^y^	5.3 ± 0.1 ^b^	18.4 ± 0.2 ^y^
Linoleic	7.2	4.0 ± 0.1 ^b^	16.0 ± 0.0 ^z^	4.6 ± 0.1 ^a^	13.7 ± 0.4 ^y^	3.8 ± 0.1 ^b^	13.7 ± 0.1 ^y^
α-linolenic	3.3	1.7 ± 0.0 ^b^	7.2 ± 0.1 ^z^	2.0 ± 0.0 ^a^	6.2 ± 0.1 ^y^	1.7 ± 0.0 ^b^	6.2 ± 0.0 ^y^
**FFA**	**Raw FFA**	**P4** **Hexane 1:10**		**P5** **Hexane 1:15**		**P6** **Hexane 1:20**	
		**SF**	**LF**	**SF**	**LF**	**SF**	**LF**
Lauric	50.3	64.6 ± 0.2 ^a^	33.0 ± 0.3 ^w^	64.3 ± 0.0 ^a^	42.3 ± 0.1 ^y^	65.0 ± 0.3 ^a^	45.3 ± 0.2 ^z^
Myristic	10.3	10.8 ± 0.0 ^b^	9.8 ± 0.1 ^y^	11.2 ± 0.0 ^a^	9.7 ± 0.1 ^y^	10.9 ± 0.0 ^b^	10.1 ± 0.0 ^z^
Palmitic	13.2	13.9 ± 0.1 ^c^	4.6 ± 0.0 ^wv^	15.2 ± 0.0 ^b^	6.1 ± 0.1 ^y^	16.3 ± 0.1 ^a^	6.8 ± 0.0 ^z^
Stearic	2.4	1.9 ± 0.0 ^c^	1.0 ± 0.0 ^y^	2.1 ± 0.0 ^b^	1.0 ± 0.0 ^y^	2.2 ± 0.0 ^a^	1.1 ± 0.0 ^z^
Oleic	8.6	2.9 ± 0.2 ^c^	18.9 ± 0.0 ^y^	2.4 ± 0.0 ^d^	15.1 ± 0.1 ^x^	1.8 ± 0.0 ^e^	13.4 ± 0.0 ^w^
Linoleic	7.2	2.0 ± 0.1 ^c^	14.1 ± 0.0 ^y^	1.6 ± 0.0 ^d^	11.2 ± 0.1 ^x^	1.1 ± 0.0 ^e^	9.9 ± 0.0 ^w^
α-linolenic	3.3	0.9 ± 0.1 ^c^	6.3 ± 0.0 ^y^	0.6 ± 0.0 ^cd^	5.0 ± 0.0 ^x^	0.5 ± 0.0 ^d^	4.5 ± 0.0 ^w^

Different letters in the same row (a–e for solid fraction, SF; and v–z for liquid fraction, LF) indicate statistical differences with *p* < 0.05. Minor fatty acids contributing to the total fatty acid composition were as follows: C8:0, C10:0, C14:1, C16:1, C17:0, C17:1, C18:2, C18:3 γ, C20:1, and C20:2.

**Table 3 insects-16-00171-t003:** FFA recovery (%) from hexane winterization processes at different oil-to-solvent ratios (*w*/*v*).

FFA	P1 Hexane 1:4		P2 Hexane 1:6		P3 Hexane 1:8	
	SF	LF	SF	LF	SF	LF
Lauric	91.3 ± 0.5 ^a^	8.7 ± 0.5 ^v^	89.9 ± 1.1 ^a^	10.1 ± 1.1 ^v^	85.4 ± 0.0 ^b^	14.6 ± 0.0 ^w^
Myristic	87.0 ± 0.4 ^a^	13.1 ± 0.4 ^v^	87.9 ± 1.0 ^a^	12.2 ± 1.1 ^v^	82.2 ± 0.0 ^b^	17.9 ± 0.1 ^w^
Palmitic	94.0 ± 0.3 ^a^	6.0 ± 0.3 ^v^	92.6 ± 1.1 ^a^	7.5 ± 1.1 ^v^	90.0 ± 0.2 ^b^	10.0 ± 0.1 ^w^
Stearic	91.0 ± 0.3 ^a^	9.1 ± 0.2 ^v^	91.4 ± 1.0 ^a^	8.6 ± 1.0 ^v^	88.0 ± 0.2 ^b^	12.1 ± 0.2 ^w^
Oleic	57.0 ± 1.4 ^b^	43.1 ± 1.3 ^v^	66.1 ± 1.8 ^a^	33.9 ± 1.8 ^u^	50.5 ± 1.3 ^c^	49.5 ± 1.3 ^w^
Linoleic	55.8 ± 1.4 ^b^	44.3 ± 1.5 ^v^	65.2 ± 1.8 ^a^	34.8 ± 1.8 ^u^	49.2 ± 1.3 ^c^	50.9 ± 1.3 ^w^
α-linolenic	55.2 ± 1.2 ^b^	44.9 ± 1.2 ^v^	64.6 ± 2.2 ^a^	35.5 ± 2.2 ^u^	48.4 ± 1.3 ^c^	51.6 ± 1.3 ^w^
Total FFA	85.2 ± 2.6	16.8 ± 0.2	86.4 ± 2.2	15.7 ± 1.1	79.6 ± 0.2	22.7 ± 0.4
**FFA**	**P4** **Hexane 1:10**		**P5** **Hexane 1:15**		**P6** **Hexane 1:20**	
	**SF**	**LF**	**SF**	**LF**	**SF**	**LF**
Lauric	81.2 ± 0.1 ^c^	18.8 ± 0.1 ^x^	64.8 ± 0.8 ^d^	35.3 ± 0.8 ^y^	52.8 ± 0.2 ^e^	47.2 ± 0.1 ^z^
Myristic	71.0 ± 0.2 ^c^	29.1 ± 0.2 ^x^	58.2 ± 1.0 ^d^	41.8 ± 1.0 ^y^	45.6 ± 0.3 ^e^	54.4 ± 0.3 ^z^
Palmitic	87.0 ± 0.1 ^c^	13.0 ± 0.1 ^x^	75.3 ± 0.8 ^d^	24.8 ± 0.8 ^y^	65.2 ± 0.1 ^e^	34.9 ± 0.1 ^z^
Stearic	80.6 ± 0.9 ^c^	19.4 ± 0.8 ^x^	71.5 ± 0.9 ^d^	28.6 ± 0.9 ^y^	60.1 ± 0.3 ^e^	39.9 ± 0.3 ^z^
Oleic	25.3 ± 1.1 ^d^	74.7 ± 1.1 ^x^	16.1 ± 0.4 ^e^	84.0 ± 0.5 ^y^	9.4 ± 0.1 ^f^	90.7 ± 0.1 ^z^
Linoleic	24.3 ± 0.5 ^d^	75.7 ± 0.4 ^x^	14.4 ± 0.3 ^e^	85.7 ± 0.4 ^y^	8.0 ± 0.1 ^f^	92.0 ± 0.1 ^z^
α-linolenic	22.9 ± 2.1 ^d^	77.1 ± 2.1 ^x^	13.4 ± 0.3 ^e^	86.6 ± 0.4 ^y^	7.5 ± 0.0 ^f^	92.5 ± 0.0 ^z^
Total FFA	70.5 ± 0.8	32.0 ± 0.4	54.2 ± 0.4	44.9 ± 1.1	43.2 ± 0.4	55.4 ± 0.1

Different letters in the same row (a–f for solid fraction, SF; and u–z for liquid fraction, LF) indicate statistical differences with *p* < 0.05. Minor fatty acids contributing to the total fatty acid composition were as follows: C8:0, C10:0, C14:1, C16:1, C17:0, C17:1, C18:2, C18:3 γ, C20:1, and C20:2.

**Table 4 insects-16-00171-t004:** Peroxide value (mEq O_2_/kg) of hexane winterization processes at different oil-to-solvent ratios (*w*/*v*).

	SF	LF
P1	0.78 ± 0.18 *	2.01 ± 0.30 ^b^ *
P2	1.11 ± 0.04 *	3.42 ± 0.07 ^ab^ *
P3	1.06 ± 0.11 *	3.60 ± 0.48 ^a^ *
P4	1.41 ± 0.47	1.99 ± 0.32 ^b^
P5	1.25 ± 0.20	2.60 ± 0.57 ^ab^
P6	1.84 ± 0.39	2.83 ± 0.33 ^ab^

Different letters in the same column indicate statistical differences with *p* < 0.05 among different processes (no statistical differences were detected in solid fraction, SF; and a–b for liquid fraction, LF). * indicates statistical differences with *p* < 0.05 among SF and LF.

**Table 5 insects-16-00171-t005:** FFA concentration (wt%) and recovery (%) from winterization processes with various solvents at a 1:10 oil-to-solvent ratio (*w*/*v*).

CONCENTRATION (wt%)												
FFA	Raw FFA	Hexane				Ethanol				Acetone		
		SF		LF		SF		LF		SF		LF
Lauric	50.3	64.6 ± 0.2 ^a^		33.0 ± 0.3		31.7 ± 1.5		59.1 ± 0.4 ^b^		56.3 ± 0.1 ^c^		50.8 ± 0.1
Myristic	10.3	10.8 ± 0.0 ^b^		9.8 ± 0.1		13.9 ± 0.6		9.8 ± 0.3 ^c^		12.0 ± 0.0 ^a^		8.8 ± 0.0
Palmitic	13.2	13.9 ± 0.1 ^b^		4.6 ± 0.0		36.7 ± 1.8		5.4 ± 0.5 ^c^		17.4 ± 0.1 ^a^		3.7 ± 0.0
Stearic	2.4	1.9 ± 0.0 ^b^		1.0 ± 0.0		5.4 ± 0.1		0.8 ± 0.1 ^c^		2.5 ± 0.0 ^a^		0.6 ± 0.0
Oleic	8.6	2.9 ± 0.2 ^c^		18.9 ± 0.0		3.8 ± 0.0		9.2 ± 0.1 ^a^		3.9 ± 0.0 ^b^		12.9 ± 0.1
Linoleic	7.2	2.0 ± 0.1 ^c^		14.1 ± 0.0		2.4 ± 0.0		6.5 ± 0.2 ^a^		2.7 ± 0.0 ^b^		9.6 ± 0.1
α-linolenic	3.3	0.9 ± 0.1 ^c^		6.3 ± 0.0		1.1 ± 0.0		2.6 ± 0.1 ^a^		1.2 ± 0.0 ^b^		4.3 ± 0.0
**RECOVERY (%)**												
**FFA**	**Hexane**				**Ethanol**				**Acetone**			
	**SF**		**LF**		**SF**		**LF**		**SF**		**LF**	
Lauric	81.2 ± 0.1 ^a^		18.8 ± 0.1		16.3 ± 2.0		83.7 ± 2.0 ^a^		55.2 ± 0.6 ^b^		44.8 ± 0.6	
Myristic	71.0 ± 0.2 ^a^		29.1 ± 0.2		33.9 ± 3.9		66.1 ± 3.9 ^ab^		60.3 ± 0.3 ^b^		39.7 ± 0.3	
Palmitic	87.0 ± 0.1 ^a^		13.0 ± 0.1		71.2 ± 3.1		28.8 ± 3.1 ^b^		84.0 ± 0.3 ^a^		16.0 ± 0.3	
Stearic	80.6 ± 0.9 ^a^		19.4 ± 0.8		71.6 ± 4.1		28.4 ± 4.1 ^b^		81.4 ± 0.1 ^a^		18.6 ± 0.1	
Oleic	25.3 ± 1.1 ^b^		74.7 ± 1.1		13.0 ± 1.0		87.0 ± 1.0 ^a^		25.1 ± 0.5 ^b^		74.9 ± 0.5	
Linoleic	24.3 ± 0.5 ^b^		75.7 ± 0.4		11.7 ± 0.8		88.3 ± 0.8 ^a^		24.0 ± 0.0 ^b^		76.0 ± 0.0	
α-linolenic	22.9 ± 2.1 ^b^		77.1 ± 2.1		12.8 ± 0.7		87.2 ± 0.7 ^a^		23.7 ± 0.1 ^b^		76.3 ± 0.1	
Total FFA	70.5 ± 0.8		32.0 ± 0.4		28.1 ± 2.5		77.6 ± 1.2		51.7 ± 0.3		46.4 ± 0.6	

Comparison was performed between the fractions in which lauric acid was enriched. Different letters in the same row indicate statistical differences with *p* < 0.05. SF: solid fraction; LF: liquid fraction. Minor fatty acids contributing to the total fatty acid composition were as follows: C8:0, C10:0, C14:1, C16:1, C17:0, C17:1, C18:2, C18:3 γ, C20:1, and C20:2.

**Table 6 insects-16-00171-t006:** Optimal operating conditions for lauric acid separation by distillation obtained by COSMO/Aspen process simulations.

		Temperature (°C)	Pressure (mbar)	Purity (%)	Recovery (%)
One-stage flash distillation	Case 1 (max purity)	102	0.11	85.8	20.2
	Case 2 (max recovery)	134	0.27	53.4	98.9
Multistage distillation (10-stage)		120	0.19	96.7	80.0

**Table 7 insects-16-00171-t007:** Optimal operating conditions for ethyl laurate acid separation by distillation obtained by COSMO/Aspen process simulations.

		Temperature (°C)	Pressure (mbar)	Purity (%)	Recovery (%)
One-stage flash distillation	Case 1 (max purity)	82	0.18	87.1	26.2
	Case 2 (max recovery)	112	0.25	54.9	98.9
Multistage distillation (10-stage)		120	1.10	97.4	80.0

## Data Availability

The raw data supporting the conclusions of this article will be made available by the authors on request.

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
