# Peer review of "Assessment of Scalable Fractionation Methodologies to Produce Concentrated Lauric Acid from Black Soldier Fly (*Hermetia illucens*) Larvae Fat"

_insects, 2025, doi:10.3390/insects16020171_

Round 1

Reviewer 1 Report

Comments and Suggestions for Authors

Review of insects-3352771-v2

Such an interesting manuscript, but it is not a match for this journal, Insects. This manuscript is more suitable to Processes than to Insects. ASPEN, COSMO, supercritical CO2 extraction, reboiler, multi-stage distillation, are of chemical engineering process design, and not related to insects. Yes the subject of interest is fat from black soilder fly (Hermetia illucens), but this manuscript does not discuss anything about the aspect of entomology, insect science at all. Please submit to Processes.

Other comments:

1.       What is winterization? Show a schematic diagram to illustrate it. What is the difference with scCO2 extraction and multi-stage distillation?

2.       With critical point of CO2 at around 31 °C, and 73.8 bar, this technology is interesting because of the ability to extract heat-sensitive materials at near room temperature. However, in line 203, why did you use the operating parameters of 55 °C and 115 bar (almost twice the temperature, and 1.5× the pressure)? Yes it is still in the region of supercritical fluid in the phase diagram of CO2, but will it be safe and not degrade the fatty acids from BSFL fat? What are the guarantees or assurance that 55 °C and 115 bar CO2 will perform better than 31 °C and 73.8 bar CO2?

3.       Figure 2a and 2b: Of course multiple-stage distillation with 10 trays will perform much better than that of single stage, because of more contact between the feed and the solvent. This manuscript will be more interesting if the ASPEN simulation is utilized for determining the optimum stage number, instead of simply comparing 1 stage vs 10 stages. So, is it 3 stages, 12 stages, or what is the optimum stage number?

4.       What is the maximum capacity of the pilot scale scCO2 extractor? Handling 200 mL/h fatty acid is not a pilot scale (line 197).

5.       Line 181: A 280 cm (twice the height of a teenager) supercritical CO2 reactor handling 73.8 bar CO2 (>70 times atmospheric pressure) has a thickness of SS316 stainless steel of how many cm? With internal diameter of less than 3 cm, it is more of a pipe instead of a reactor (height >90× of the diameter). A pipe with working volume less than 2 L (gross), to be exact. That kind of working volume is not a pilot scale.  

6.       Line 141: “Transesterification reactions of BSFL fat”… --> do you mean production of biodiesel? Why bother fractionating biodiesel to be lauric acid etc. (as shown in Table 2)?

7.       Line 141: Why fatty acid ETHYL ester (FAEE), and not METHYL ester (FAME)? How to ensure the product of your reaction is ETHYL ester, not METHYL ester? Write down in the manuscript about the advantages of FAME vs FAEE.

8.       Production of FAME/biodiesel from BSFL is not that remarkable from the perspective of yield (Sustainability 2022, 14, 13993). The yield looks very good (>90%, (1) Environ. Chem. Lett. 2019, 17, 1143–115; (2) Bioresour. Technol. 2015, 194, 276–282; (4) Energy Conv. Manag. 2018, 158, 168–175) if and only if it is compared against dried BSFL.  The yield definitely will drop significantly if normalized to wet BSFL, especially when compared to the mass of the feed of BSF, which is only 14.36 g per kg waste (chicken manure and rapeseed straw, J. Clean. Prod. 2020, 263, 121495). So, why bother fractionating a low yield product? How economically feasible will your process(es) go?

9.       Again, this manuscript does not suitable for Insects, and it is more to Processes (note: many fundamental issues to address, e.g. process design, process parameters, engineering economy).

Author Response

Response to Reviewer 1 Comments

Review of insects-3352771-v2

Such an interesting manuscript, but it is not a match for this journal, Insects. This manuscript is more suitable to Processes than to Insects. ASPEN, COSMO, supercritical CO2 extraction, reboiler, multi-stage distillation, are of chemical engineering process design, and not related to insects. Yes the subject of interest is fat from black soilder fly (Hermetia illucens), but this manuscript does not discuss anything about the aspect of entomology, insect science at all. Please submit to Processes.

Response: Thank you very much for taking the time to review this manuscript. Please find the detailed responses below and the corresponding revisions/corrections highlighted/in track changes in the re-submitted files. Regarding the topic of interest, please refer to point 9.

Other comments:

  1. What is winterization? Show a schematic diagram to illustrate it. What is the difference with scCO2 extraction and multi-stage distillation?

Response: The term winterization refers to a low-temperature crystallization process, as explained in lines 90-91 of the revised manuscript. Similar to the processes of scCO2 extraction and multi-stage distillation, its goal is to fractionate a sample to concentrate some of its components. The difference between the three processes lies in the physicochemical variable on which the fractionation is based. In scCO2 extraction, the fractionation occurs due to differences in the solubility of the sample components in scCO2, whereas in multi-stage distillation, the components are fractionated based on their differences in volatility (boiling point). In contrast, winterization processes rely on differences in melting points to fractionate the sample components, which, in the case of fatty acids, is primarily determined by the number of unsaturations, as mentioned in lines 91-94 of the revised manuscript.

A simplified diagram of a winterization process is attached. However, due to the high number of figures included in the manuscript, we believe it is more appropriate not to include this in the article, as it is also a fairly simple process.

  1. With critical point of CO2at around 31 °C, and 73.8 bar, this technology is interesting because of the ability to extract heat-sensitive materials at near room temperature. However, in line 203, why did you use the operating parameters of 55 °C and 115 bar (almost twice the temperature, and 1.5× the pressure)? Yes it is still in the region of supercritical fluid in the phase diagram of CO2, but will it be safe and not degrade the fatty acids from BSFL fat? What are the guarantees or assurance that 55 °C and 115 bar CO2 will perform better than 31 °C and 73.8 bar CO2?

Response: Supercritical fluid extraction allows the modulation of fluid density and, therefore, its solvent capacity through pressure and temperature variables. In this way, the density at 31 °C and 73.8 bar is different from that at 55 °C and 115 bar. As mentioned in lines 205-207 of the revised manuscript, the selection of the optimal pressure and temperature was based on preliminary studies. This previous approach was conducted with coconut fats due to their similar fatty acid composition, but it was not included in the manuscript to make it more easily understandable for the reader.

It should be noted that the conditions of 31 °C and 73.8 bar correspond to the critical point of CO₂, meaning the lowest conditions at which the supercritical region is reached. However, it is common for the optimal extraction of compounds from different matrices to occur at higher temperatures and, particularly, higher pressures. Moreover, the conditions of 55 °C and 115 bar are still mild and do not affect the stability of the fatty acid ethyl esters.

  1. Figure 2a and 2b: Of course multiple-stage distillation with 10 trays will perform much better than that of single stage, because of more contact between the feed and the solvent. This manuscript will be more interesting if the ASPEN simulation is utilized for determining the optimum stage number, instead of simply comparing 1 stage vs 10 stages. So, is it 3 stages, 12 stages, or what is the optimum stage number?

Response: The present study focused experimentally on the processes of winterization and supercritical CO₂ extraction to concentrate lauric acid. However, the aim was to include distillation processes through theoretical simulation to compare the results obtained using different fractionation technologies. The distillation study was designed for two distinct scenarios: one-stage and 10-stage, and using fatty acids in both their free form and as ethyl esters. In our opinion, the observed differences help determine the concentration and recovery of lauric acid under varying scenarios. Intermediate situations of lauric acid concentration and recovery could be evaluated through simulations with an intermediate number of stages. However, the goal was to evaluate distillation under very different conditions and compare the results with those from winterization and supercritical CO₂ extraction.

  1. What is the maximum capacity of the pilot scale scCO2extractor? Handling 200 mL/h fatty acid is not a pilot scale (line 197).

Response: In this case, the location where the supercritical extraction takes place cannot be considered an extraction cell but rather an extraction column used for processing liquid samples. As explained in section 2.3.2, Supercritical Fluid Extraction of FAEE, in this study, the supercritical extraction consists of a countercurrent extraction where CO₂ is pumped from the bottom of the column, and the liquid sample is introduced from the top. According to this design, although the extraction column has a specific internal volume, this cannot be regarded as the maximum sample capacity of the equipment. In such cases, the liquid sample flow rate and its relationship with the CO₂ flow rate are used as key parameters.

Laboratory-scale SFE equipment is generally designed for solid samples, which are introduced into small-volume extraction cells (mL scale). The use of a countercurrent extraction column with a liquid sample pump handling flows of 100–1000 L requires components (valves, separators, CO₂ pump, CO₂ condenser, etc.) on a larger scale. In our opinion, this setup falls within the scope of pilot plant-scale operations.

  1. Line 181: A 280 cm (twice the height of a teenager) supercritical CO2reactor handling 73.8 bar CO2 (>70 times atmospheric pressure) has a thickness of SS316 stainless steel of how many cm? With internal diameter of less than 3 cm, it is more of a pipe instead of a reactor (height >90× of the diameter). A pipe with working volume less than 2 L (gross), to be exact. That kind of working volume is not a pilot scale.  

Response: As mentioned in the previous section, the extraction takes place neither in a cell nor in a pipe; instead, it occurs in an extraction column where CO₂ is pumped from the bottom and the liquid sample from the top. Therefore, the extraction occurs in a countercurrent flow. Due to the CO₂ and liquid sample flow rates used, as well as the dimensions of both the column and the other components of the supercritical extraction equipment, we consider the equipment to be of pilot plant scale.

  1. Line 141: “Transesterification reactions of BSFL fat”… --> do you mean production of biodiesel? Why bother fractionating biodiesel to be lauric acid etc. (as shown in Table 2)?

Response: Yes, for SFE processes the starting material was biodiesel (FAEE) instead of free fatty acids (FFA). As explained in lines 199-204 of the revised manuscript, BSFL ethanolysis instead of hydrolysis was selected to produce the starting SFE material because: (i) FAEE are in a liquid state, easily pumpable, and thus capable of flowing through the extraction column, unlike FFA, which are in a solid state at ambient temperature; and (ii) FAEE are less polar than FFA, making them more soluble in super-critical CO2, allowing the process to be carried out under milder conditions.

  1. Line 141: Why fatty acid ETHYL ester (FAEE), and not METHYL ester (FAME)? How to ensure the product of your reaction is ETHYL ester, not METHYL ester? Write down in the manuscript about the advantages of FAME vs FAEE.

Response: We fully agree with the reviewer that the production of FAME would have been an equally valid option. However, we chose to transform the fatty acids into their corresponding FAEE to avoid the use of methanol, which is highly toxic and not permitted in the food industry.

According to the reviewer the sentence: “FAEE were selected instead of fatty acid methyl esters (FAME) to avoid the use of methanol, which is highly toxic and not permitted in the food industry” has been included in the manuscript” (lines 204-205 of the revised manuscript).

  1. Production of FAME/biodiesel from BSFL is not that remarkable from the perspective of yield (Sustainability202214, 13993). The yield looks very good (>90%, (1)  Chem. Lett. 201917, 1143–115; (2) Bioresour. Technol. 2015194, 276–282; (4) Energy Conv. Manag. 2018158, 168–175) if and only if it is compared against dried BSFL.  The yield definitely will drop significantly if normalized to wet BSFL, especially when compared to the mass of the feed of BSF, which is only 14.36 g per kg waste (chicken manure and rapeseed straw, J. Clean. Prod. 2020263, 121495). So, why bother fractionating a low yield product? How economically feasible will your process(es) go?

Response: The present study is part of a research line focused on using Hermetia illucens larvae as an alternative and much more sustainable protein source in terms of land use, water consumption, global warming, etc., compared to other types of livestock, for use in aquaculture feed. These larvae undergo a process of slaughtering, drying, and defatting to obtain the protein source (1). In this process, a fat fraction is generated as a co-product, accounting for approximately 40% of the total dry larva. Of this 40%, 50% is lauric acid. Therefore, this study aims to add value to this fat fraction, which is already being generated as an integral part of the process and, in our opinion, offers a very interesting yield.

(1) Hurtado-Ribeira, R. et al. (2023). Evaluation of the interrelated effects of slaughtering, drying, and defatting methods on the composition and properties of black soldier fly (Hermetia illucens) larvae fat. Current Research in Food Science, 7:100633.

  1. Again, this manuscript does not suitable for Insects, and it is more to Processes (note: many fundamental issues to address, e.g. process design, process parameters, engineering economy).

Response: We agree with the reviewer that the manuscript primarily focuses on technological processes and may be well-suited for a journal primarily dedicated to this topic. However, in our opinion, the manuscript can also fit well within Insects, as it is part of a broader research line aimed at valorizing and obtaining high-value-added compounds from insect sources. It provides an integral explanation of their potential applications and relevance from the perspective of the circular economy, particularly within the context of the emerging field of insect research.

Reviewer 2 Report

Comments and Suggestions for Authors

This paper addresses an important and timely topic: the development of scalable and sustainable methodologies for concentrating lauric acid from Black Soldier Fly larvae (Hermetia illucens) fat. The work is well-positioned within the growing interest in circular economy principles and the valorization of alternative resources in the food and chemical industries. By focusing on three fractionation techniques—winterization, supercritical fluid extraction (SFE), and distillation—the authors aim to provide insights into methods for producing high-purity lauric acid that can serve various industries, from food to cosmetics and pharmaceuticals.

The research is promising and contributes to the broader goals of sustainability and circular economy. However, to enhance its academic rigor and practical relevance, the authors should: Clearly articulate the novelty and impact of the work and provide a balanced interpretation of the results, considering trade-offs in scalability and environmental impact; expand the literature review and improve clarity in presentation.

Below is a detailed critique of its weaknesses and suggestions for improvement.

- The research gap is not clearly articulated. While the authors identify the importance of lauric acid concentration, they do not provide strong quantitative or comparative arguments to highlight the limitations of current methods or the specific advantages of using Black Soldier Fly larvae fat.

- Include a detailed comparison of lauric acid yields and production efficiencies from traditional sources (e.g., coconut oil, palm kernel oil) versus Black Soldier Fly larvae. Provide a stronger justification for the focus on these larvae as a sustainable alternative.

- “Lauric acid exhibits various beneficial properties, leading to its extensive use in industries such as pharmaceuticals, cosmetics, animal feed, and food products [5–7].” Please, can you add more specific references? I suggest https://doi.org/10.3390/insects13010041 and https://doi.org/10.1007/s00253-024-13005-9

- The conclusions are optimistic but lack balance. The advantages of supercritical fluid extraction are emphasized without adequately discussing its higher energy requirements or CO2 consumption. The challenges of high-vacuum operation in distillation are mentioned only briefly.

- Include a more balanced discussion that explicitly highlights trade-offs such as costs, energy use, and environmental impacts for each technique. Consider adding a cost-benefit analysis to provide practical context for scalability and applicability.

- The review of existing methodologies is limited in scope: alternative solvents and bio-based approaches are not sufficiently discussed. Recent advancements in lipid extraction and lauric acid production from non-traditional sources are overlooked.

- Some sections are overly dense and difficult to follow, particularly the technical descriptions of experimental setups and simulations. Figures and tables lack intuitive annotations and explanatory captions.

- Revise dense sentences and eliminate redundancies for improved readability. Ensure that all visual elements have clear labels, legends, and contextual explanations to guide readers effectively.

- While the study emphasizes environmental sustainability, it does not include quantitative metrics such as energy consumption, CO2 emissions, or waste reduction. Incorporating these metrics would significantly strengthen the paper’s alignment with green chemistry principles.

Author Response

Response to Reviewer 2 Comments

This paper addresses an important and timely topic: the development of scalable and sustainable methodologies for concentrating lauric acid from Black Soldier Fly larvae (Hermetia illucens) fat. The work is well-positioned within the growing interest in circular economy principles and the valorization of alternative resources in the food and chemical industries. By focusing on three fractionation techniques—winterization, supercritical fluid extraction (SFE), and distillation—the authors aim to provide insights into methods for producing high-purity lauric acid that can serve various industries, from food to cosmetics and pharmaceuticals.

The research is promising and contributes to the broader goals of sustainability and circular economy. However, to enhance its academic rigor and practical relevance, the authors should: Clearly articulate the novelty and impact of the work and provide a balanced interpretation of the results, considering trade-offs in scalability and environmental impact; expand the literature review and improve clarity in presentation.

Response: Thank you very much for taking the time to review this manuscript. Please find the detailed responses below and the corresponding revisions/corrections highlighted/in track changes in the re-submitted files.

Below is a detailed critique of its weaknesses and suggestions for improvement.

- The research gap is not clearly articulated. While the authors identify the importance of lauric acid concentration, they do not provide strong quantitative or comparative arguments to highlight the limitations of current methods or the specific advantages of using Black Soldier Fly larvae fat.

Response: The primary objective of this study is not to highlight the technological limitations of current methods for concentrating lauric acid but rather to compare different technologies for this purpose, with a particular emphasis on those aligned with green chemistry (the experimental part of the study). Thus, the study is primarily focused on determining the lauric acid concentration and yield values achieved through these technologies and making a comparison between them.

On the other hand, the present work does not aim to demonstrate that Black Soldier Fly fat is economically more advantageous than other sources such as coconut or palm kernel oil. Instead, it seeks to add value to a fat that is being produced in an emerging market, specifically the insect industry. It should be noted that this study is part of a research line focused on using Hermetia illucens larvae as an alternative and much more sustainable protein source in terms of land use, water consumption, global warming, etc., compared to other types of livestock, for use in aquaculture feed. In this process, a fat fraction is generated as a co-product, accounting for approximately 40% of the total dry larva. Therefore, this study aims to add value to this fat fraction, which is already being generated as an integral part of the process and, in our opinion, offers a very interesting applicability.

- Include a detailed comparison of lauric acid yields and production efficiencies from traditional sources (e.g., coconut oil, palm kernel oil) versus Black Soldier Fly larvae. Provide a stronger justification for the focus on these larvae as a sustainable alternative.

Response: In our opinion, this point has already been addressed in the previous section. This study does not aim to justify the use of Black Soldier Fly larvae fat over other traditional sources, such as coconut oil or palm kernel oil, but rather to add value to the fat from these larvae, which is already being generated as a co-product in the production of protein derivatives primarily intended for animal feed (aquaculture).

- “Lauric acid exhibits various beneficial properties, leading to its extensive use in industries such as pharmaceuticals, cosmetics, animal feed, and food products [5–7].” Please, can you add more specific references? I suggest https://doi.org/10.3390/insects13010041 and https://doi.org/10.1007/s00253-024-13005-9

Response: We agree with the reviewer and have therefore replaced the following references, as we believe they were not appropriately selected.

Mortensen, A.; Aguilar, F.; Crebelli, R.; Di Domenico, A.; Dusemund, B.; Frutos, M.J.; Galtier, P.; Gott, D.; Gundert‐Remy, U.; Leblanc, J.; et al. Re‐evaluation of Fatty Acids (E 570) as a Food Additive. EFSA J. 2017, 15, doi:10.2903/J.EFSA.2017.4785.

Ullah, S.; Zhang, J.; Xu, B.; Tegomo, A.F.; Sagada, G.; Zheng, L.; Wang, L.; Shao, Q. Effect of Dietary Supplementation of Lauric Acid on Growth Performance, Antioxidative Capacity, Intestinal Development and Gut Microbiota on Black Sea Bream (Acanthopagrus schlegelii). PLoS One 2022, 17, doi:10.1371/JOURNAL.PONE.0262427.

Instead, we have included the references suggested by the reviewer:

  1. Franco, A.; Scieuzo, C.; Salvia, R.; Pucciarelli, V.; Borrelli, L.; Addeo, N.F.; Bovera, F.; Laginestra, A.; Schmitt, E.; Falabella, P. Antimicrobial Activity of Lipids Extracted from Hermetia illucens Reared on Different Substrates. Appl Microbiol Biotechnol 2024, 108, doi.org/10.1007/s00253-024-13005-9.
  2. Franco, A.; Salvia, R.; Scieuzo, C.; Schmitt, E.; Russo, A.; Falabella, P. Lipids from Insects in Cosmetics and for Personal Care Products. Insects 2022, 13, 41, doi.org/10.3390/insects13010041.

- The conclusions are optimistic but lack balance. The advantages of supercritical fluid extraction are emphasized without adequately discussing its higher energy requirements or CO2 consumption. The challenges of high-vacuum operation in distillation are mentioned only briefly.

Response: According to the reviewer the conclusions have been tuned down. The following sentence has been included: “Nevertheless, selecting the most appropriate technology for concentrating lauric acid would require conducting a comprehensive study to thoroughly assess the economic costs, including equipment investment, energy consumption, and environmental impacts, in addition to evaluate specific challenges associated with each technology.”

In addition, a paragraph mentioning costs, energy requirements, or CO2 consumption has been included in the final part of the Results and Discussion section. However, in our opinion, a comprehensive study regarding economic parameters would be very interesting but should be complementary and conducted in a future investigation.

- Include a more balanced discussion that explicitly highlights trade-offs such as costs, energy use, and environmental impacts for each technique. Consider adding a cost-benefit analysis to provide practical context for scalability and applicability.

Response: We have included a paragraph at the end of the Results and Discussion section that broadly addresses some aspects mentioned by the reviewer. The paragraph is as follows:

“At this point, a further study focused on the costs, energy consumption, and environmental impacts associated with the three technologies would be necessary. Winterization is cost-effective and relatively low in energy consumption, making it suitable for small to medium-scale operations. However, its reliance on solvents raises environmental and safety concerns due to disposal challenges [49]. Supercritical CO2 extraction provides good concentration and selectivity, making it ideal for high-value applications, but its initial cost can be high due to the need for advanced equipment, along with CO2 consumption, which, while often recyclable, still may have a notable environmental footprint [50]. Vacuum distillation, on the other hand, is highly scalable and efficient for large volumes, with lower thermal degradation of products due to reduced boiling points under vacuum. However, its energy demands for maintaining high vacuum conditions and heating are substantial, leading to higher operational costs and potential environmental impacts [51].”

  1. Kreulen, H. P. Fractionation and Winterization of Edible Fats and Oils. J. Am. Oil Chem. Soc. 1976, 53, 393-396, doi: 10.1007/BF02605729.
  2. Nikolai, P.; Rabiyat, B.; Aslan A.; Ilmutdin, A. Supercritical CO2: Properties and Technological Applications - A Review. J. Therm. Sci. 2019, 28, 394-430, doi: 10.1007/s11630-019-1118-4.
  3. Atta, M. S.; Khan, H.; Ali, M.; Tariq, R.; Yasir, A.U.; Iqbal, M.M.; Din, S.U.; Krzywanski, J. Simulation of Vacuum Distillation Unit in Oil Refinery: Operational Strategies for Optimal Yield Efficiency. Energies 2024, 17, 3806, doi:10.3390/en17153806.

However, in our opinion, a more extensive discussion on this matter may be speculative. As mentioned in previous sections, we believe that a comprehensive and in-depth study of economic parameters would be highly valuable, but it should be considered as a complementary effort and undertaken in a future investigation.

- The review of existing methodologies is limited in scope: alternative solvents and bio-based approaches are not sufficiently discussed. Recent advancements in lipid extraction and lauric acid production from non-traditional sources are overlooked.

Response: According to the reviewer the next paragraph has been added to the Introduction section:

“In recent years, the combination of technologies classified as 'green', such as enzymatic pretreatments, ultrasound, pressurized liquid extraction, microwave, as well as SFE and winterization, has undergone significant advancements for the extraction and fractionation of lipid components from different matrices [27-29]. Furthermore, there is currently a growing trend in the use of non-traditional, bio-based or food-grade solvents in these applications, such as limonene [30].”

  1. Picot-Allain, C., Mahomoodally, M.F.; Ak, G.; Zengin, G. Conventional versus green extraction techniques — a comparative perspective. Curr. Opin. Food Sci. 2021, 40, 144-156, doi:10.1016/j.cofs.2021.02.009
  2. Kumar, M.; Barbhai, M.D.; Puranik, S.; Radha; Natta, S.; Senapathy, M.; Dhumal, S.; Singh, S.; Kumar, S.; V.P.; Deshmukh, V.P.; et al. Combination of green extraction techniques and smart solvents for bioactives recovery. TrAC Trends Anal. Chem. 2023, 169, 117286, doi:10.1016/j.trac.2023.117286
  3. Amiri-Rigi, A.; Abbasi, S.; Scanlon, M.G. Enhanced lycopene extraction from tomato industrial waste using microemulsion technique: optimization of enzymatic and ultrasound pre-treatments. Innovat. Food Sci. Emerg. Technol. 2016, 35, 160-167, doi:10.1016/j.ifset.2016.05.004
  4. Chemat, S.; Tomao, V.; Chemat, F. Limonene as Green Solvent for Extraction of Natural Products. In Green Solvents I. Mohammad, A., Inamuddin, Eds.; Springer, Dordrecht, Netherlands, 2012, pp. 175-187, doi:10.1007/978-94-007-1712-1

- Some sections are overly dense and difficult to follow, particularly the technical descriptions of experimental setups and simulations. Figures and tables lack intuitive annotations and explanatory captions.

Response: We consider all tables and figures have their corresponding captions and annotations for proper understanding. Including additional annotations, in our opinion, would make them denser and can hinder their comprehension.

- Revise dense sentences and eliminate redundancies for improved readability. Ensure that all visual elements have clear labels, legends, and contextual explanations to guide readers effectively.

Response: The manuscript has been thoroughly revised, and corrections have been made in accordance with the suggestions provided by the reviewers.

- While the study emphasizes environmental sustainability, it does not include quantitative metrics such as energy consumption, CO2 emissions, or waste reduction. Incorporating these metrics would significantly strengthen the paper’s alignment with green chemistry principles.

Response: We fully agree with the reviewer that an economic study would be very interesting and would complement the work conducted. However, as explained in the previous sections, we want to emphasize that the present study focuses on the process of concentrating lauric acid from H. illucens fat, evaluating the responses of concentration and recovery. The study is centered on the evaluation of these responses through the three technologies studied. For this reason, we believe that an economic study addressing energy consumption, CO2 emissions, or waste reduction should be carried out as a subsequent and complementary investigation. Otherwise, in our opinion, the work would become excessively extensive.

Reviewer 3 Report

Comments and Suggestions for Authors

This study explored various fractionation methodologies to concentrate lauric acid and maximize its potential.Through this work, it has been demonstrated that winterization and supercritical  fluid extraction are effective in concentrating lauric acid from the fat of Hermetia illucens larvae.Additionally, the experimental results obtained were compared with flash and multistage distillation processes simulation, a more conventional method of concentration of fatty acids.. Overall, the employed methodologies proved highly efficient in concentrating lauric acid, yielding a product of commercial interest and high added value.The study has a reasonable design, rich experimental content, and reliable results, which provides a very high reference and value for the extraction and application of lauric acid. It is recommended for acceptance after minor revision.

1. Line 26,115,116 et al.:(2023) should be deleted.

2. Line 539: Hermetia Illucens should be italicized.

3. Line 566: Hermetia Illucens should be italicized.

Author Response

Response to Reviewer 3 Comments

This study explored various fractionation methodologies to concentrate lauric acid and maximize its potential. Through this work, it has been demonstrated that winterization and supercritical  fluid extraction are effective in concentrating lauric acid from the fat of Hermetia illucens larvae. Additionally, the experimental results obtained were compared with flash and multistage distillation processes simulation, a more conventional method of concentration of fatty acids. Overall, the employed methodologies proved highly efficient in concentrating lauric acid, yielding a product of commercial interest and high added value. The study has a reasonable design, rich experimental content, and reliable results, which provides a very high reference and value for the extraction and application of lauric acid. It is recommended for acceptance after minor revision.

Response: Thank you very much for taking the time to review this manuscript. Please find the detailed responses below and the corresponding revisions/corrections highlighted/in track changes in the re-submitted files.

  1. Line 26,115,116 et al.:”(2023)” should be deleted.

Response: According to the reviewer, the year has been deleted in all references included in the body of the manuscript.

  1. Line 539: “Hermetia Illucens” should be italicized.

Response: According to the reviewer the term has been corrected.

  1. Line 566: “Hermetia Illucens” should be italicized.

Response: According to the reviewer the term has been corrected.

Reviewer 4 Report

Comments and Suggestions for Authors

The paper, Assessment of Scalable Fractionation Methodologies to Produce Concentrated Lauric Acid from Black Soldier Fly (Hermetia illucens) Larvae Fat, is well-structured and provides valuable insights.

However, the discussion section does not sufficiently analyze or contextualize the results. A deeper exploration of their significance, potential limitations, and implications for future research or industrial applications is missing. Please, expand the discussion to critically evaluate the findings, compare methodologies, and outline the broader relevance of the work to the field of sustainable lipid production and insect-based bioproducts. This addition will enhance the scientific depth of the paper. Lastly, it is important to ensure that the text complements the data presented in the tables rather than repeating it.

In my opinion, it will be suitable for publication in Insects only after major revisions.

Line 15: “the fat content of Hermetia illucens larvae"

Line 23: "was required as a previous step" (missing article "a")

Line 29: replace “by process simulation" with “using process simulation”

Line 29: "demonstrating that using this methodology it is possible" could be rephrased to "demonstrating that this methodology can achieve"

Lines 30-31: The sentence "but high vacuum is required to prevent the product thermal degradation" could be rephrased to "but a high vacuum is required to prevent thermal degradation of the product" for clearer and more correct phrasing.

INTRODUCTION

Line 45: "which constitutes approximately 50% of the total fatty acids" or "making it attractive due to its high lauric acid content, which accounts for approximately 50% of the total fatty acids."

Line 46: Lauric acid a saturated fatty acid containing twelve carbon atoms, …"

Lines 60-64: "In animal feeding, interest in lauric acid has increased in recent years, as a growth enhancer, and due to its immunomodulatory and protective effects at the gastrointestinal level, which are related to microbiota modulation".

Line 67: "it has not yet been explored".

Line 82: "As an example, Vázquez & Akoh (2010)".

Lines 93-95: "Supercritical fluid extraction (SFE) has advanced significantly since its inception and is widely recognized as a clean and environmentally friendly 'green' processing technique."

Line 95: it would be better to not start a sentence whit an abbreviation/sigle/acronym, please replace “CO2” with “carbon dioxide”

Lines 102-104: The word "Besides" is informal. You can use "Furthermore" or "Moreover": "Furthermore, fatty acids released in their ester form are more stable than their corresponding free fatty acids and are preferred for supercritical fractionation due to their higher solubility in dense CO2."

Line 104: "To the best of our knowledge, the use of sc-CO2 has been limited".

MATERIALS AND METHODS

Line 116: please add a comma “Potassium hydroxide, ethanol (96% v/v), and absolute ethanol were obtained from Scharlab S.L. (Barcelona, Spain).”

Line 116: please specify the menaing of “v/v” the first time you use it, and “w/v” too (line 120) as well as “w/w”…

Lines 121-125: seems there are some repetition (methyl alcol and methanol are the same thing, right?). Please check

Lines 122: please specify the meaning of HPLC the first time you use it

Lines 133-134: "After 20 min, the reaction was stopped by adding 40 mL of distilled water, followed by 65 mL of 4 M hydrochloric acid to lower the pH to 2 and release the free fatty acids (FFA)."

Lines 141-142: "Transesterification reactions of BSFL fat were carried out to convert the triacylglycerols into their corresponding fatty acid ethyl esters (FAEE)."

Lines 160-161: Suggestion: "w/v" should be clarified as "weight/volume" to avoid ambiguity. Also, the ratios might be written as "1:4, 1:6, 1:8, 1:10, 1:15, 1:20 (w/v)" for readability.

Lines 161-162: "At the optimal FFA to solvent ratio, two additional experiments were conducted using absolute ethanol and acetone as solvents for comparison."

Line 171: "Concentration or purity: weight percentage of a single fatty acid in the composition."

Line 173: "Recovery: % of one fatty acid regarding its amount in the starting material."

Line 178: replace “….. extractor (model: SF2000, Thar Technology, Pitts-178 burgh, PA, USA) …” (now it is similar to the one at line 190. Please, use “m” and not “M”.

Line 181: "counter-current"

Line 195: it is better not to start with an abbreviation “Sc-CO2”, please change it. Check throught he entire text for others!

Lines 198-202: it might benefit from additional clarification or reference to literature supporting this assertion.

Lines 206-207: "Products were collected from the top (extract) and the bottom (raffinate) of the column."

Line 256: AOAC stands for “Association of Official Analytical Collaboration“? Please add this (same for AOCS.

Lines 271-273: "To evaluate the impact of the processes on lipid oxidation, a key attribute of lipid quality, the peroxide value (PV)..."

RESULTS AND DISCUSSION

Line 292: "Lauric Acid Concentration via Winterization."

Line 294: Replace "ca." with "approximately."

Lines 310, 327, 363, 398, 423, …: Use italicized p values: "p < 0.05" (in tables too)

Line 330: Replace "increasement" with "increase."

Line 364: Replace "stating FFA" with "starting FFA."

Lines 343: The abbreviation "LF" (Liquid Fraction) and “SF” (solid fraction) are introduced without definition in the text (they are present in the tables). Add the full term before its first use, e.g., "liquid fraction (LF)."

Line 400: mass flow rate right?

Lines 408-411: The mass balance accuracy is stated as "around 90%," but there is no explanation for the 10% discrepancy. This could be due to measurement errors, losses, or unaccounted material. Add a sentence explaining the potential reasons for the 10% discrepancy in the mass balance.

Lines 426-428: Rephrase to "Primary oxidation compounds, which capture active oxygen species, become more polar and less soluble in CO2, enriching them in the raffinate."

Lines 433-434: "Given that distillation is a widely used technique..."

Line 459: The term "thermal stability" could be clarified. Does it refer to avoiding degradation or maintaining a consistent boiling point? Provide context.

Section 3.4 Comparison of results from the three fractionation technologies: Summarize comparisons concisely in one sentence instead of repeating individual results for each method. Maybe a table instead of a figure (Figure 6) with concentration and recovery of lauric acid by the different technologies assessed would be clearer. In my opinion there is a need of specify what causes the variability in recovery varies depending on conditions or operational parameters (e.g., operating conditions like pressure or temperature). Please, add a final sentence explaining why SFE is considered a compromise (e.g., balance between concentration and recovery compared to other methods).

Line 488: "To improve lauric acid purity beyond SFE results, a 10-stage distillation would be necessary."

CONCLUSIONS

Lines 497-498: "Winterization primarily separated components based on the presence of double bonds, whereas sc-CO2 separated components by chain length."

Lines 502-503: This statement is unsupported. Add a brief justification for why winterization is cost-effective and simpler (e.g., lower equipment and operational costs) in the discussion section.

Line 505: “…, highlighting this process as a viable and effective method for obtaining high-value lipids from edible insects."

Author Response

Response to Reviewer 4 Comments

The paper, Assessment of Scalable Fractionation Methodologies to Produce Concentrated Lauric Acid from Black Soldier Fly (Hermetia illucens) Larvae Fat, is well-structured and provides valuable insights.

However, the discussion section does not sufficiently analyze or contextualize the results. A deeper exploration of their significance, potential limitations, and implications for future research or industrial applications is missing. Please, expand the discussion to critically evaluate the findings, compare methodologies, and outline the broader relevance of the work to the field of sustainable lipid production and insect-based bioproducts. This addition will enhance the scientific depth of the paper. Lastly, it is important to ensure that the text complements the data presented in the tables rather than repeating it.

In my opinion, it will be suitable for publication in Insects only after major revisions.

Response: Thank you very much for taking the time to extensively review this manuscript. Please find the detailed responses below and the corresponding revisions/corrections highlighted/in track changes in the re-submitted files.

Line 15: “the fat content of Hermetia illucens larvae"

            Response: The sentence has been corrected.

Line 23: "was required as a previous step" (missing article "a")

            Response: The article “a” has been included.

Line 29: replace “by process simulation" with “using process simulation”

            Response: The sentence has been changed.

Line 29: "demonstrating that using this methodology it is possible" could be rephrased to "demonstrating that this methodology can achieve"

            Response: The sentence has been rephrased.

Lines 30-31: The sentence "but high vacuum is required to prevent the product thermal degradation" could be rephrased to "but a high vacuum is required to prevent thermal degradation of the product" for clearer and more correct phrasing.

            Response: The sentence has been rephrased.

INTRODUCTION

Line 45: "which constitutes approximately 50% of the total fatty acids" or "making it attractive due to its high lauric acid content, which accounts for approximately 50% of the total fatty acids."

Response: The sentence has been replaced with “making it attractive due to its high lauric acid content, which accounts for approximately 50% of the total fatty acids”.

Line 46: Lauric acid a saturated fatty acid containing twelve carbon atoms, …"

            Response: The sentence has been rephrased.

Lines 60-64: "In animal feeding, interest in lauric acid has increased in recent years, as a growth enhancer, and due to its immunomodulatory and protective effects at the gastrointestinal level, which are related to microbiota modulation".

Response: The sentence has been revised in accordance with the reviewer's suggestion.

Line 67: "it has not yet been explored".

            Response: The sentence has been corrected.

Line 82: "As an example, Vázquez & Akoh (2010)".

            Response: The sentence has been corrected.

Lines 93-95: "Supercritical fluid extraction (SFE) has advanced significantly since its inception and is widely recognized as a clean and environmentally friendly 'green' processing technique."

Response: The sentence has been revised in accordance with the reviewer's suggestion.

Line 95: it would be better to not start a sentence whit an abbreviation/sigle/acronym, please replace “CO2” with “carbon dioxide”

            Response: The term has been changed.

Lines 102-104: The word "Besides" is informal. You can use "Furthermore" or "Moreover": "Furthermore, fatty acids released in their ester form are more stable than their corresponding free fatty acids and are preferred for supercritical fractionation due to their higher solubility in dense CO2."

Response: The sentence has been revised in accordance with the reviewer's suggestion.

Line 104: "To the best of our knowledge, the use of sc-CO2 has been limited".

Response: The sentence has been revised in accordance with the reviewer's suggestion.

MATERIALS AND METHODS

Line 116: please add a comma “Potassium hydroxide, ethanol (96% v/v), and absolute ethanol were obtained from Scharlab S.L. (Barcelona, Spain).”

            Response: The sentence has been corrected.

Line 116: please specify the menaing of “v/v” the first time you use it, and “w/v” too (line 120) as well as “w/w”…

Response: According to the reviewer, the meanings of “weight/weight” (w/w), “weight/volume” (w/v), and “volume/volume” (v/v) were specified the first time they appeared in the text.

Lines 121-125: seems there are some repetition (methyl alcol and methanol are the same thing, right?). Please check

Response: We agree with the reviewer and have corrected all the information with the following sentence: “Hexane (95%), acetone, and methyl alcohol anhydrous, all HPLC grade solvents, were purchased from Macron Fine Chemicals™ (Center Valley, PA, USA).”

Lines 122: please specify the meaning of HPLC the first time you use it

Response: In our opinion, the term "HPLC grade" should remain as it is, because it is category (purity grade). Furthermore, it appears in the text only once. The meaning of HPLC is widely known, and in our view, the term "High-Performance Liquid Chromatography grade" would not be meaningful and it sounds very strange.

Lines 133-134: "After 20 min, the reaction was stopped by adding 40 mL of distilled water, followed by 65 mL of 4 M hydrochloric acid to lower the pH to 2 and release the free fatty acids (FFA)."

Response: The sentence has been revised in accordance with the reviewer's suggestion. However, since the term "FFA" was already defined, the abbreviation has been used.

Lines 141-142: "Transesterification reactions of BSFL fat were carried out to convert the triacylglycerols into their corresponding fatty acid ethyl esters (FAEE)."

Response: The sentence has been revised in accordance with the reviewer's suggestion.

Lines 160-161: Suggestion: "w/v" should be clarified as "weight/volume" to avoid ambiguity. Also, the ratios might be written as "1:4, 1:6, 1:8, 1:10, 1:15, 1:20 (w/v)" for readability.

Response: According to the reviewer, the explanation "weight/volume (w/v)" has been included in the text. In the tables, the ratios were already denoted as "1:4, 1:6, 1:8, 1:10, 1:15, 1:20 (w/v)." In the discussion, using the notation P1–P6 improves readability, in our opinion.

Lines 161-162: "At the optimal FFA to solvent ratio, two additional experiments were conducted using absolute ethanol and acetone as solvents for comparison."

Response: The sentence has been revised in accordance with the reviewer's suggestion.

Line 171: "Concentration or purity: weight percentage of a single fatty acid in the composition."

Response: The sentence has been revised in accordance with the reviewer's suggestion.

Line 173: "Recovery: % of one fatty acid regarding its amount in the starting material."

Response: The sentence has been revised in accordance with the reviewer's suggestion.

Line 178: replace “….. extractor (model: SF2000, Thar Technology, Pitts-178 burgh, PA, USA) …” (now it is similar to the one at line 190. Please, use “m” and not “M”.

Response: According to the reviewer, the information in both paragraphs has been corrected.

Line 181: "counter-current"

            Response: The term has been corrected.

Line 195: it is better not to start with an abbreviation “Sc-CO2”, please change it. Check throught he entire text for others!

Response: The term has been corrected.

Lines 198-202: it might benefit from additional clarification or reference to literature supporting this assertion.

Response: This paragraph mentions the differences in melting point and polarity between the FFA and FAEE forms. In our opinion, these are basic physicochemical characteristics of the compounds that do not require support from specific bibliographic sources.

Lines 206-207: "Products were collected from the top (extract) and the bottom (raffinate) of the column."

Response: The sentence has been revised in accordance with the reviewer's suggestion.

Line 256: AOAC stands for “Association of Official Analytical Collaboration“? Please add this (same for AOCS).

Response: “Association of Official Agricultural Chemists (AOAC)” and “American Oil Chemists' Society (AOCS)” have been added to the text.

Lines 271-273: "To evaluate the impact of the processes on lipid oxidation, a key attribute of lipid quality, the peroxide value (PV)..."

Response: The sentence has been revised in accordance with the reviewer's suggestion.

RESULTS AND DISCUSSION

Line 292: "Lauric Acid Concentration via Winterization."

Response: The titles have been revised in accordance with the reviewer's suggestion.

Line 294: Replace "ca." with "approximately."

            Response: "ca." has been replaced with "approximately" in all text.

Lines 310, 327, 363, 398, 423, …: Use italicized p values: "p < 0.05" (in tables too)

            Response:p” has been italicized in all text and tables.

Line 330: Replace "increasement" with "increase."

            Response: The term has been corrected.

Line 364: Replace "stating FFA" with "starting FFA."

            Response: The term has been corrected.

Lines 343: The abbreviation "LF" (Liquid Fraction) and “SF” (solid fraction) are introduced without definition in the text (they are present in the tables). Add the full term before its first use, e.g., "liquid fraction (LF)."

            Response: The abbreviation has been added at its first use in the text.

Line 400: mass flow rate right?

Response: The term has been corrected.

Lines 408-411: The mass balance accuracy is stated as "around 90%," but there is no explanation for the 10% discrepancy. This could be due to measurement errors, losses, or unaccounted material. Add a sentence explaining the potential reasons for the 10% discrepancy in the mass balance.

Response: The explanation “Approximately 10% of the material remained as liquid impregnating the packing within the extraction column” has been included in the manuscript.

Lines 426-428: Rephrase to "Primary oxidation compounds, which capture active oxygen species, become more polar and less soluble in CO2, enriching them in the raffinate."

            Response: The sentence has been rephrased.

Lines 433-434: "Given that distillation is a widely used technique..."

Response: The sentence has been revised in accordance with the reviewer's suggestion.

Line 459: The term "thermal stability" could be clarified. Does it refer to avoiding degradation or maintaining a consistent boiling point? Provide context.

Response: The information has been clarified by a correction in the sentence: “Table 6 collects the design of 10-stage distillation proposed in this work to concentrate lauric acid, avoiding thermal degradation by operating at 120 ºC in the reboiler.”

Section 3.4 Comparison of results from the three fractionation technologies: Summarize comparisons concisely in one sentence instead of repeating individual results for each method. Maybe a table instead of a figure (Figure 6) with concentration and recovery of lauric acid by the different technologies assessed would be clearer. In my opinion there is a need of specify what causes the variability in recovery varies depending on conditions or operational parameters (e.g., operating conditions like pressure or temperature). Please, add a final sentence explaining why SFE is considered a compromise (e.g., balance between concentration and recovery compared to other methods).

Response: The idea of including the summary of results as Figure 6 was to enable a quick and intuitive visualization of the comparison of results obtained through the different methodologies. For this reason, we firmly believe that this summary should appear as a figure, especially since, in the rest of the manuscript, the results are primarily presented through tables.

According to the reviewer the following explanation has been added to discussion (lines 416-421 in the revised version):

“Thus, it was confirmed that the selected combination of parameters—temperature (55 ºC) and pressure (115 bar)—resulted in a CO2 density with sufficient solvent capacity to effectively and selectively solubilize lauric acid. Furthermore, the solvent-to-feed ratio employed ensured an adequate amount of CO2 to extract this compound with high re-covery, which constitutes a significant proportion of the starting material.”

In addition, the following sentence has been revised and completed (lines 495-498 in the revised version):

“As illustrated in Figure 6, SFE represents the most suitable compromise between concentration (~80%) and recovery (~85%) of lauric acid, as it provided an optimal balance between these responses in comparison to other methodologies evaluated.”

Line 488: "To improve lauric acid purity beyond SFE results, a 10-stage distillation would be necessary."

Response: The sentence has been revised in accordance with the reviewer's suggestion.

 CONCLUSIONS

Lines 497-498: "Winterization primarily separated components based on the presence of double bonds, whereas sc-CO2 separated components by chain length."

Response: The sentence has been revised in accordance with the reviewer's suggestion.

Lines 502-503: This statement is unsupported. Add a brief justification for why winterization is cost-effective and simpler (e.g., lower equipment and operational costs) in the discussion section.

Response: We have included a paragraph at the end of the Results and Discussion section that broadly addresses some aspects mentioned by the reviewer. The paragraph is as follows:

“At this point, a further study focused on the costs, energy consumption, and environmental impacts associated with the three technologies would be necessary. Winterization is cost-effective and relatively low in energy consumption, making it suitable for small to medium-scale operations. However, its reliance on solvents raises environmental and safety concerns due to disposal challenges [49]. Supercritical CO2 extraction provides good concentration and selectivity, making it ideal for high-value applications, but its initial cost can be high due to the need for advanced equipment, along with CO2 consumption, which, while often recyclable, still may have a notable environmental footprint [50]. Vacuum distillation, on the other hand, is highly scalable and efficient for large volumes, with lower thermal degradation of products due to reduced boiling points under vacuum. However, its energy demands for maintaining high vacuum conditions and heating are substantial, leading to higher operational costs and potential environmental impacts [51].”

  1. Kreulen, H. P. Fractionation and Winterization of Edible Fats and Oils. J. Am. Oil Chem. Soc. 1976, 53, 393-396, doi: 10.1007/BF02605729.
  2. Nikolai, P.; Rabiyat, B.; Aslan A.; Ilmutdin, A. Supercritical CO2: Properties and Technological Applications - A Review. J. Therm. Sci. 2019, 28, 394-430, doi: 10.1007/s11630-019-1118-4.
  3. Atta, M. S.; Khan, H.; Ali, M.; Tariq, R.; Yasir, A.U.; Iqbal, M.M.; Din, S.U.; Krzywanski, J. Simulation of Vacuum Distillation Unit in Oil Refinery: Operational Strategies for Optimal Yield Efficiency. Energies 2024, 17, 3806, doi:10.3390/en17153806.

However, in our opinion, a more extensive discussion on this matter may be speculative. We believe that a comprehensive and in-depth study of economic parameters would be highly valuable, but it should be considered as a complementary effort and undertaken in a future investigation.

Accordingly, the following sentence has been added to the end of Conclusions section:

“Nevertheless, selecting the most appropriate technology for concentrating lauric acid would require conducting a comprehensive study to thoroughly assess the economic costs, including equipment investment, energy consumption, and environmental impacts, in addition to evaluate specific challenges associated with each technology.”

Line 505: “…, highlighting this process as a viable and effective method for obtaining high-value lipids from edible insects."

Response: The sentence has been revised in accordance with the reviewer's suggestion.

Round 2

Reviewer 1 Report

Comments and Suggestions for Authors

Review of insects-3352771

1.       If no optimum stage number is shown here, then why bother use ASPEN or maybe other process simulation software (e.g. HYSYS)?

2.       I am not convinced 200 mL/h (a glass of water per hour??) is a pilot scale. Show the photo of your equipments, along with scale bar or metering tape.

3.       Show the equipments that you claimed can hande flows of 100-1000 L. Nevertheless, do you really have 100 L fatty acid (resulting from how many kg BSF larvae harvested over how many months)? If not, then no, your equipments are not a pilot plant, just a mere lab scale equipments.

4.       If you insist that the reaction takes place in extraction column (and not in extraction cell), then show the photo of the extraction column, and write the details of that column dimension.

5.       A 280 cm (twice the height of a teenager) supercritical CO2reactor handling 73.8 bar CO2 (>70 times atmospheric pressure) has a thickness of SS316 stainless steel of how many cm? --> this question was not addressed in your point-by-point response.

6.       If your BSFL contains 40% fat (or dry larvae), then please show what kind of feed they were fed of? Is it really consistent 40% fat, or not? Add error bar (standard deviation) of this value.

7.       Again, this manuscript does not suitable for Insects, and it is more to Processes (note: many fundamental issues to address, e.g. process design, process parameters, engineering economy). If you insist on submitting this manuscript discussing technological process to Insects, with the reason of “part of a broader research line”, then show the chemical engineering economy analysis (initial cost, operational cost, NPV, CNPV, ROI, BEP, etc.).

Author Response

Responses to Reviewer 1 Comments (second revision)

Review of insects-3352771

  1. If no optimum stage number is shown here, then why bother use ASPEN or maybe other process simulation software (e.g. HYSYS)?

Response: The primary objective of this study was to examine the concentration and recovery of lauric acid from Hermetia illucens fat using winterization and supercritical fluid extraction (SFE) technologies. However, given that distillation is a widely employed industrial method for the fractionation of fatty acids, a comparative analysis with this technique was also undertaken. A theoretical simulation was performed to estimate the expected outcomes under two distinct scenarios: (a) a single-stage distillation process and (b) a multi-stage distillation process with ten stages. This study was intended solely as a reference for comparison with winterization and SFE. A comprehensive assessment of the trade-offs between lauric acid concentration and recovery across a range of distillation stages (from 1 to 10) would have required conducting ten separate theoretical simulations, which was beyond the primary scope of this research.

  1. I am not convinced 200 mL/h (a glass of water per hour??) is a pilot scale. Show the photo of your equipments, along with scale bar or metering tape.

Response: We agree with the reviewer that the pilot scale can be considered larger than our equipment. However, we believe that our equipment has larger dimensions than those of a lab-scale system. Therefore, we consider "semi-pilot scale" to be the most appropriate term. Accordingly, this term has replaced “pilot scale” in the revised version of the manuscript. We have also attached photos of a lab-scale SFE system and the semi-pilot scale SFE equipment used in this study.

Lab-scale SFE equipment:

Lab-scale 1

Lab-scale 2

Semi-pilot scale SFE column:

Semi-pilot scale

Semi-pilot scale SFE cell and separators:

Semi-pilot scale

  1. Show the equipments that you claimed can hande flows of 100-1000 L. Nevertheless, do you really have 100 L fatty acid (resulting from how many kg BSF larvae harvested over how many months)? If not, then no, your equipments are not a pilot plant, just a mere lab scale equipments.

Response: In the previous section, photos of the semi-pilot scale SFE equipment used in the study are shown. As described in section 2.3.2 of the manuscript, the product obtained from BSFL ethanolysis (FAEE mixture) was pumped from the top at a constant flow rate of 2.9 g/min (approximately 200 mL/h) for 60 minutes. Thus, approximately 200 mL of product was pumped in each experiment.

The dried BSF larvae were provided by Entomo Agroindustrial (Cehegín, Murcia, Spain) after being blanched for slaughtering and subsequently oven-dried. The fat was then extracted by mechanical pressing. Multiple extractions were performed to obtain a sufficient amount of fat for the subsequent steps of the study.

In the next step, chemical ethanolysis was conducted to obtain FAEE from the extracted fat before fractionation by SFE. Several transesterification reactions, each using 500 g of BSFL fat, were carried out to generate a sufficient amount of starting material for all the SFE processes.

  1. If you insist that the reaction takes place in extraction column (and not in extraction cell), then show the photo of the extraction column, and write the details of that column dimension.

Response: The following image presents the supercritical extraction column utilized in this study.

Semi pilot-scale column

As described in section 2.3.2 of the manuscript, “The experimental device comprises a counter-current extraction column (280 cm total height) with two levels of feed inlet (oil sample) and two separators (500 mL each) where the decompression and recovery of the extract takes place (Figure 1). Each section of the packed column and the separators has independent control of temperature (± 2 ºC). The extraction column (316 stainless) has an internal diameter of 2.97 cm, and it is packed with a structured material (17-4PH H1150).”

  1. A 280 cm (twice the height of a teenager) supercritical CO2 reactor handling 73.8 bar CO2 (>70 times atmospheric pressure) has a thickness of SS316 stainless steel of how many cm? --> this question was not addressed in your point-by-point response.

Response: Unfortunately, the thickness of the stainless steel in Thar Designs´ supercritical extraction columns is not publicly specified. However, these devices are designed to withstand high-pressure conditions while adhering to the required regulations and safety standards.

  1. If your BSFL contains 40% fat (or dry larvae), then please show what kind of feed they were fed of? Is it really consistent 40% fat, or not? Add error bar (standard deviation) of this value.

Response: The BSFL were fed a conventional diet based on wheat bran. This information has been included in the revised manuscript. The manuscript does not explicitly mention the total fat content in BSFL, as it can exhibit somewhat elevated variability. However, this data is not relevant for the supercritical fractionation, as a sufficient number of defatting processes were carried out to obtain enough starting material to concentrate lauric acid. On the other hand, it is specified that lauric acid accounts for approximately 50% of the total fatty acids present in the fat, which is relevant for the study.

  1. Again, this manuscript does not suitable for Insects, and it is more to Processes (note: many fundamental issues to address, e.g. process design, process parameters, engineering economy). If you insist on submitting this manuscript discussing technological process to Insects, with the reason of “part of a broader research line”, then show the chemical engineering economy analysis (initial cost, operational cost, NPV, CNPV, ROI, BEP, etc.).

Response: We appreciate the reviewer's feedback regarding the suitability of our manuscript for Insects. While we acknowledge the importance of chemical engineering economic analysis in process design, the primary focus of our study is the evaluation of scalable methodologies for lauric acid concentration from Hermetia illucens fat, rather than a detailed techno-economic assessment.

Our research aligns with the scope of Insects as it explores novel approaches to valorizing insect-derived lipids, which is relevant to the journal’s readership. We understand that a full economic evaluation, including parameters such as initial investment, operational costs, and financial indicators, would provide additional insights; however, conducting such an extensive analysis would excessively lengthen the manuscript and go beyond the intended scope of this study.

Given that our objective is to assess the feasibility and effectiveness of different fractionation techniques rather than optimize them from an economic standpoint, we respectfully maintain our focus on the technological aspects. We hope the reviewer understands this rationale and that our manuscript remains suitable for consideration within the journal’s framework.

Reviewer 4 Report

Comments and Suggestions for Authors

Just two small revisions of formatting and style:

Line 87: it is in bold, please correct it “ments for the extraction and fractionation of lipid components from different matrices”

Line 417: replace “55 ºC” with “55 °C”  

Author Response

Responses to Reviewer 4 Comments (second revision)

Review of insects-3352771

We appreciate the reviewer’s corrections and suggestions.

Line 87: it is in bold, please correct it “ments for the extraction and fractionation of lipid components from different matrices”

Response: Currently, the sentence is not in bold.

Line 417: replace “55 ºC” with “55 °C”  

Response: This has been corrected.

Round 3

Reviewer 1 Report

Comments and Suggestions for Authors

Review of insects-3352771-v4

OK, make sure to include the figures of the equipments in the Supplementary Information.